# The global human impact on biodiversity

François Keck[1,2 ✉], Tianna Peller[1,2], Roman Alther[1,2], Cécilia Barouillet[3], Rosetta Blackman[1,2], Eric Capo[4], Teofana Chonova[5], Marjorie Couton[1,2], Lena Fehlinger[6], Dominik Kirschner[2,7,8], Mara Knüsel[1,2], Lucile Muneret[9,10], Rebecca Oester[1,2,11], Kálmán Tapolczai[12,13], Heng Zhang[1,2] & Florian Altermatt[1,2 ✉]

Human activities drive a wide range of environmental pressures, including habitat change, pollution and climate change, resulting in unprecedented effects on biodiversity[1,2]. However, despite decades of research, generalizations on the dimensions and extent of human impacts on biodiversity remain ambiguous. Mixed views persist on the trajectory of biodiversity at the local scale[3] and even more so on the biotic homogenization of biodiversity across space[4,5]. We compiled 2,133 publications covering 97,783 impacted and reference sites, creating an unparalleled dataset of 3,667 independent comparisons of biodiversity impacts across all main organismal groups, habitats and the five most predominant human pressures[1,6]. For all comparisons, we quantified three key measures of biodiversity to assess how these human pressures drive homogenization and shifts in composition of biological communities across space and changes in local diversity, respectively. We show that human pressures distinctly shift community composition and decrease local diversity across terrestrial, freshwater and marine ecosystems. Yet, contrary to long-standing expectations, there is no clear general homogenization of communities. Critically, the direction and magnitude of biodiversity changes vary across pressures, organisms and scales at which they are studied. Our exhaustive global analysis reveals the general impact and key mediating factors of human pressures on biodiversity and can benchmark conservation strategies.

Biodiversity change poses a critical threat to human societies from local to global scales, highlighting the urgent need for understanding the complex relationship between human pressures and their effects on ecosystems[1]. Human pressures, broadly classified in five main types—land-use change, resource exploitation, pollution, climate change and invasive species[6]—can enhance or reduce species diversity locally. Crucially, by impacting biodiversity at local scales, effects of human pressures can similarly impact biodiversity patterns among communities at broader spatial scales. This includes shifts in species composition among biological communities across a region, as well as increases and decreases in similarity between communities (homogenization and differentiation, respectively).

Despite decades of accumulating evidence of human impacts on biological communities, the trajectory of biodiversity in the Anthropocene remains unclear and attempts for syntheses have yielded mixed and debated results for both local diversity[7–12] and composition of communities across space[4,5,13,14]. Understanding and generalizing the impacts of human pressures on biodiversity and how they are mediated by key factors, such as the spatial scale or type of pressure, remains a major challenge. Such information is critical if we are to understand whether the actions taken to prevent further loss and change of contemporary biodiversity are successful and to give insights into appropriate strategies to monitor the success of mitigation actions.

Crucially, previous research attempting to generalize biodiversity change has hitherto neglected two key elements. First, most past studies looked at biodiversity change across time using individual time series and did not contrast findings to reference controls[9]. Second, previous studies have rarely differentiated between changes in local diversity versus changes in variation in diversity across space. Unfortunately, the studies that have integrated these elements are generally restricted to a certain type of pressure or to a particular biome[15–19]. Consequently, we lack generalizations on the effects of human pressures on ecosystems and our understanding of biodiversity change with regard to its different dimensions remains incomplete. Thus, we are limited in our capacity to disentangle its underlying drivers. Given the multifaceted aspect of biodiversity and the plurality of drivers, organisms and spatial scales, the present lack of synthetic understanding and attribution of general impacts of human pressures on biological communities is hindering adequate actions and mitigation strategies[20,21].

[1]Department of Evolutionary Biology and Environmental Studies, University of Zurich, Zurich, Switzerland. [2]Eawag, Swiss Federal Institute of Aquatic Science and Technology, Department of Aquatic Ecology, Dübendorf, Switzerland. [3]INRAE, Université Savoie Mont Blanc, CARRTEL, Thonon-les-Bains, France. [4]Department of Ecology and Environmental Science, Umeå University, Umeå, Sweden. [5]Eawag, Swiss Federal Institute of Aquatic Science and Technology, Department of Environmental Chemistry, Dübendorf, Switzerland. [6]GEA Aquatic Ecology Group, University of Vic—Central University of Catalonia, Vic, Spain. [7]Department of Environmental Systems Science, Institute of Terrestrial Ecosystems, Ecosystems and landscape evolution, ETH Zürich, Zurich, Switzerland. [8]Department of Landscape Dynamics & Ecology, Swiss Federal Research Institute WSL, Birmensdorf, Switzerland. [9]INRAE, Université Paris-Saclay, AgroParisTech, UMR Agronomie, Palaiseau, France. [10]INRAE, Agroécologie, Institut Agro, Univ. Bourgogne, Université Bourgogne Franche-Comté, Dijon, France. [11]Institute of Microbiology, University of Applied Sciences and Arts of Southern Switzerland, Mendrisio, Switzerland. [12]HUN-REN Balaton Limnological Research Institute, Tihany, Hungary. [13]National Laboratory for Water Science and Water Security, HUN-REN Balaton Limnological Research Institute, Tihany, Hungary. ✉e-mail: francois.keck@gmail.com; florian.altermatt@ieu.uzh.ch

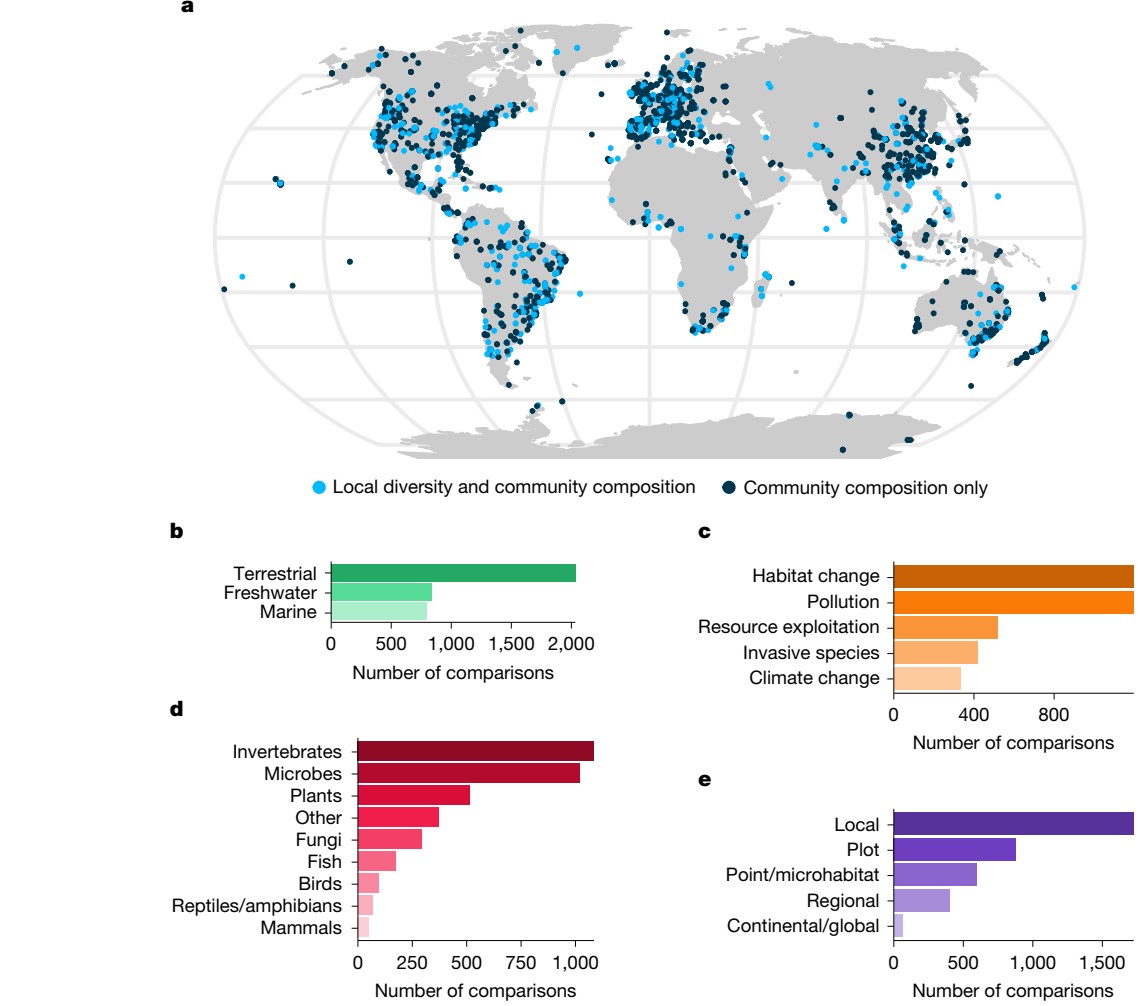

**Fig. 1 | Location of diversity comparisons and their distribution across biomes, pressures, organisms and scale. a**, Global map of the 3,667 comparisons of diversity included in this study. **b**–**e**, Distribution of comparisons of diversity by type of biome (**b**), human pressures (**c**), groups of organisms (**d**) and spatial scale (**e**). These variables correspond to the four main factors tested in this study.

Here we compiled and analysed a large dataset to assess the impacts of human pressures on biodiversity, systematically contrasting impacted versus reference communities. We first identified the global trends of community homogenization and then the associated shifts in community composition and changes in local diversity. We studied these changes through a meta-analysis of distance-based unconstrained ordination plots broadly used to assess individual and case-specific effects of human pressures on community composition. We manually and systematically extracted datapoints from these ordination plots, each of these points representing the composition of an individual biological community (Supplementary Information section 1). This meta-analytical framework, first introduced by ref. 22, allowed us to discriminate between changes of homogeneity and shifts in composition of biological communities in relation to human pressures. By contrast to previous studies mostly restricted to biomonitoring time series, we focused on direct impact studies, considering any of the five most predominant anthropogenic pressures: land-use change, resource exploitation, pollution, climate change and invasive species[1,6]. For each study included, we compared impacted communities to the reference (control) scenario. Contrary to individual biodiversity time series, this allows direct quantification and comparison of the effects of human pressures[23]. Of the included studies, 32% are experimental, directly manipulating the human pressure to a reference control, whereas the remaining 68% of the studies do this impact-comparison in pairwise observational approaches.

To study the human impacts on community diversity across space, we collected 3,667 individual comparisons involving 49,401 reference communities and 48,382 impacted communities from 2,133 published studies. This global dataset includes all main groups of organisms (including plants, tetrapods, fish, insects, microbes and fungi) and is representative of the main biomes of the Earth (marine, freshwater and terrestrial). We focused and quantified changes associated with the five dominant human pressures, across several spatial scales from local to global (Fig. 1 and Extended Data Fig. 1). For each comparison, we calculated the log-response ratio—that is, the logarithm-transformed ratio of impacted to reference values—for different components of biodiversity change. First, we evaluated if the different impacted sites are more similar or dissimilar to each other than the reference sites (homogeneity: LRR homogeneity). Then, we looked at the change in species composition between impacted and reference sites (compositional shift: LRR shift). Doing so, we quantified the relative changes of the different dimensions of biodiversity across space in a standardized way. Further, we computed the change in local diversity as the log-response ratio of local diversity (LRR local diversity). We used mixed linear models to estimate the magnitude and significance of these changes and tested the effect of four groups of factors on their variation: biome, human pressure, group of organisms and spatial scale.

Contrary to general expectations, we find no evidence of systematic biotic homogenization in response to human pressures (Fig. 2a).

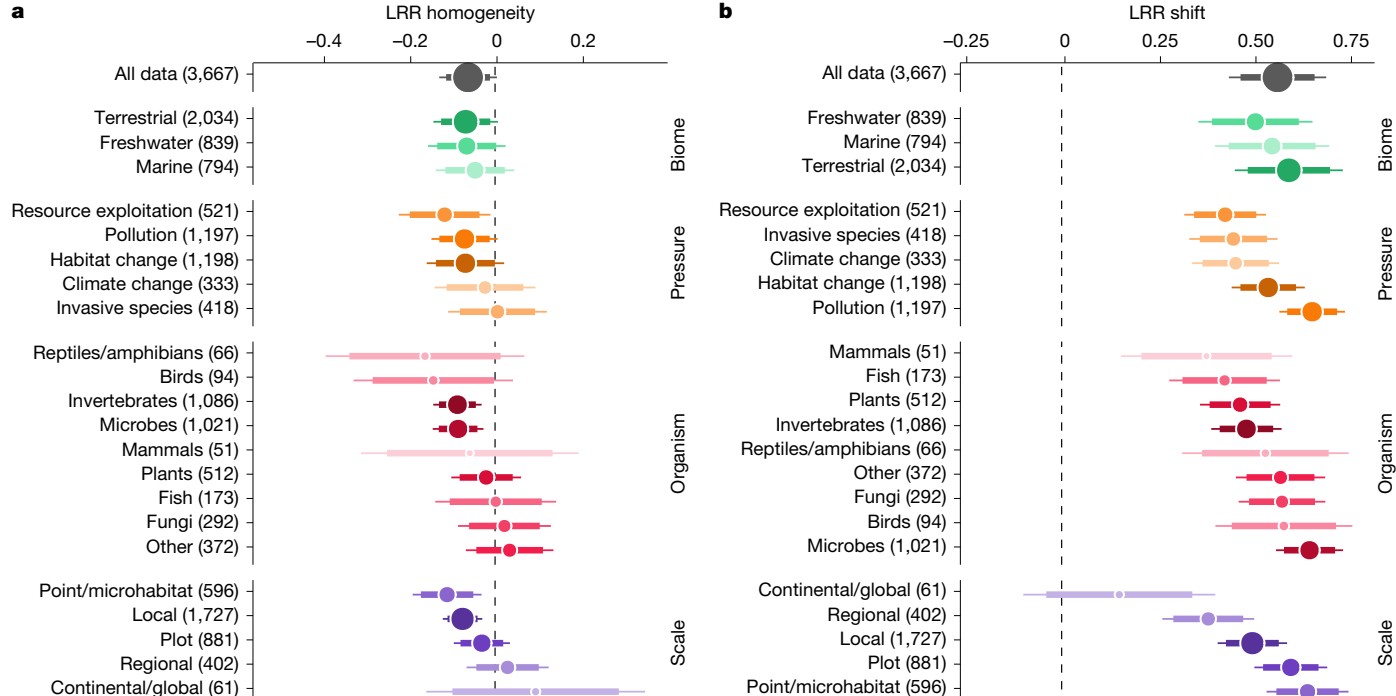

**Fig. 2 | Impacts of human pressures on homogeneity and shifts in composition of biological communities. a**, log-response ratio of community homogeneity (logarithm-transformed ratio of impacted to reference values, LRR homogeneity). **b**, log-response ratio of community composition shift (LRR shift). The global response (all data) is shown on the first row of each panel and is separated by factors in the following rows. The numbers in parentheses indicate the number of comparisons. For each category the dot represents the marginal mean computed from the model; dot size is proportional to number of studies included. The larger bar shows the 95% confidence interval and the thinner bar represents the 99% confidence interval.

The overall log-response ratio for homogeneity is close to zero, yet negative, which suggests biotic differentiation (LRR homogeneity = −0.062, 95% confidence interval (CI) = −0.012 to −0.113). Although the theory of biotic homogenization of communities under human pressure prevailed for a long time[24], recent case studies show that biotic differentiation can be regularly observed[14]. Our exhaustive meta-analysis generalizes this observation, showing that the average impact of human pressures, across all published studies, is biotic differentiation. Critically, however, we find that spatial scale significantly mediates the effects of human pressures on community homogeneity ($\chi^2$ = 10.8, $P$ = 0.029), showing from a wide range of contexts that both phenomena (biotic homogenization and differentiation) are widespread. Specifically, human pressures tend to homogenize communities at larger scales (positive LRR homogeneity; Fig. 2a) and differentiate them at smaller scales (negative LRR homogeneity). Large-scale biotic homogenization, when reported, is often linked to the redistribution of species and the facilitation of their dispersal over long distances by humans[13,25–29]. The fact that local studies show biotic differentiation can be explained by a finer sampling grain and better characterization of communities at small spatial scales, which can make biotic differentiation more apparent[14,30]. Furthermore, the well-known distance–dissimilarity relationship suggests that communities that are spatially closer are on average more similar[31], and thus more prone to differentiation than at larger scales. Finally, stochastic effects and ecological drift can promote biotic differentiation[32], and are likely to have more important roles in local impact studies in which strong pressures can completely destabilize communities by drastically reducing the number of individuals. In our systematic analysis, we indeed find a significant biotic differentiation in response to resource exploitation (LRR homogeneity = −0.117, 95% CI = −0.197 to −0.036) and pollution (LRR homogeneity = −0.071, 95% CI = −0.129 to −0.012), two types of human pressure capable of modifying ecosystems in a pronounced way over a short period of time, and thus increasing the importance of ecological drift in community assembly.

By contrast, and in line with general expectations, we find a clear shift in community composition in response to human pressures (LRR shift = 0.564, 95% CI = 0.467 to 0.661) whose magnitude varies according to the type of biome ($\chi^2$ = 12.3, $P$ = 0.002), pressure ($\chi^2$ = 42.5, $P$ < 0.001), group of organisms ($\chi^2$ = 26.1, $P$ = 0.001) and spatial scale ($\chi^2$ = 39.2, $P$ < 0.001) considered (Fig. 2b). Our analysis shows unequivocally that community composition is impacted by human pressures. Such a strong shift can be directly and consistently linked to habitat changes benefiting certain species at the cost of others through environmental filtering and niche processes. Notably, we find that all five types of human pressures (land-use change, resource exploitation, pollution, climate change and invasive species) included in our analysis significantly shift the composition of biological communities (Fig. 2b). These five ubiquitous human pressures are clearly identified in the millennium ecosystem assessment and many studies have shown how they can impact the composition of biological communities since its publication[28,33–36]. Our results show that these human pressures systematically change the composition of communities and provide critical insights on the magnitude of effects across human pressures, supporting the notion that all human pressures need to be considered when attempting to bend the curve on biodiversity loss[37]. We find that habitat change and, above all, pollution have a particularly strong effect on community composition shifts. Yet, we acknowledge that experimental studies of these two human pressures may have compared reference controls to generally relatively high treatment levels (see Extended Data Figs. 2–7 for stratified analyses separating experimental from observational data) and that ranking human pressures can be strongly context- and metric-dependent[37]. We also show significant differences in composition shifts between groups of organisms. Microbes and fungi, which contain predominantly smaller species, have the highest shifts in

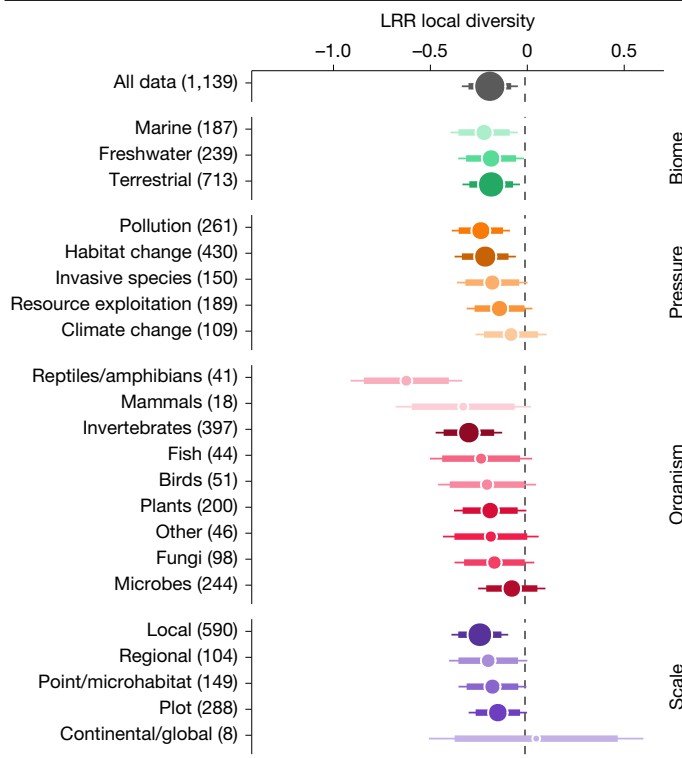

**Fig. 3 | Impacts of human pressures on local diversity.** log-response ratio of local diversity (logarithm-transformed ratio of impacted to reference values, LRR local diversity). The global response (all data) shown on the first row is separated by factors in the following rows. The numbers in parentheses indicate the number of comparisons. For each category the dot represents the marginal mean computed from the model; dot size is proportional to the number of studies included. The larger bar shows the 95% confidence interval and the thinner bar represents the 99% confidence interval.

the composition of their communities, whereas the effect is less pronounced for mammals, fish, amphibians and reptiles. It has been shown that smaller species, which generally exhibit higher diversity, shorter life cycles and higher dispersal rates relative to larger species, have higher rates of community composition change[38,39]. Here we provide evidence that this difference among groups is directly reflected in the magnitude of their response to human pressure. As for homogenization, spatial scale has an important role, with shifts in composition becoming increasingly marked as the spatial scale considered is reduced. Again, this result can be explained by better detection of rare species at finer spatial grain. However, directed shifts are also driven by the capacity of a new pool of species to colonize several impacted sites to establish similar communities, which is expected to be more likely at smaller spatial scales.

As changes in community diversity across space can be tightly coupled to changes in local diversity[40], we further examined changes in local diversity in relation to human pressures. We extracted 1,139 comparisons of local diversity (taxonomic richness) between reference and impacted communities (Fig. 1a) for a subset of 727 publications and computed for each comparison the log-response ratio for this local diversity (LRR local diversity). Overall, we find clear evidence that sites impacted by human pressures have lower local diversity (LRR local diversity = −0.181, 95% CI = −0.291 to −0.071; Fig. 3). We find that the type of pressures ($\chi^2$ = 11.3, $P$ = 0.023) and group of organisms ($\chi^2$ = 41.7, $P$ < 0.001) significantly affected local diversity change. Similarly to the results for compositional shifts, pollution and habitat change are the strongest drivers of local diversity loss. Previous syntheses are in line with our finding on the impact of land-use

change on local biodiversity[16,41,42]. However, contrary to community composition shifts, it is the largest organisms that are experiencing the strongest negative effects of human pressures for local diversity. We speculate that contemporary declines reported in vertebrate populations[1,43] may be a manifestation of these pressures, given the intrinsic link between population size and risk of local extinction. The trajectory of local diversity in the Anthropocene is the subject of intense and long-standing debate[7–10,12,44]. These studies are built on time-series analysis, which generally lack an impact–reference comparison, may be limited by the number of sites and the accuracy of the measurements[45], and must be based on adequate null model expectations[23]. We circumvent these challenges by systematically comparing impacted sites with reference sites (that is, control–impact design). In such designs, the control and impacted sets of sites are assumed to be comparable and any differences between the two treatments is attributed to a change in the impacted group relative to the reference group, which is considered as a stable baseline. Although this approach may be in some cases less sensitive than the gold-standard 'before–after control–impact' (BACI) design, which explicitly accounts for pre-existing differences between the impact and reference groups[46], it remains by far the most widely used method to measure the real and direct effect of human pressures on biological communities (more than 95% of the studies considered had a control–impact design and less than 5% had a BACI design). With this impact-focused perspective, we quantify and recall the direct and adverse effects of human pressures on local biodiversity.

Finally, our results show a link between changes in local diversity and shifts in composition and homogenization of biological communities across space. Although an interdependency of these different aspects of biodiversity is theoretically predicted[40], large-scale integrations are rare as many studies focused on one component only. We report that LRR homogeneity increases (Fig. 4a,b) and LRR shift decreases (Fig. 4c,d) with an increase of LRR local diversity ($\chi^2$ = 11.0, $P$ < 0.001 and $\chi^2$ = 42.0, $P$ < 0.001, respectively). In other words, a greater loss of species is associated with a stronger shift in composition and more differentiated communities. With a few exceptions, this pattern is highly consistent across biomes, types of pressure, groups of organisms and spatial scales (Fig. 4b,d). Inherent to any comparative study, the observed relationship does not allow us to deduce causality, yet a direct dependency between changes in local diversity, compositional shift and homogenization is not only in line with theoretical predictions, but corroborates the adverse effects of human pressures and their tangible repercussions on the various dimensions of biodiversity.

Bending the curve of contemporary biodiversity loss and change is one of the greatest challenges facing our society[47,48]. Ambitious targets have been proposed to reverse biodiversity change, yet the direction and magnitude of interdependent effects on different levels of biodiversity are still broadly debated. In particular, attributing the change in biodiversity to fundamental drivers has lagged behind[21]. By systematically assessing how the five main global human pressures impact biodiversity, we quantitatively attribute biodiversity change in impacted versus reference communities, integrating effects on both local diversity change and changes in community composition. Our comprehensive analysis provides a new and highly detailed picture of the state of knowledge available on the signal of human impacts on biodiversity and is thus an important benchmark for the development and assessment of future conservation strategies.

## Online content

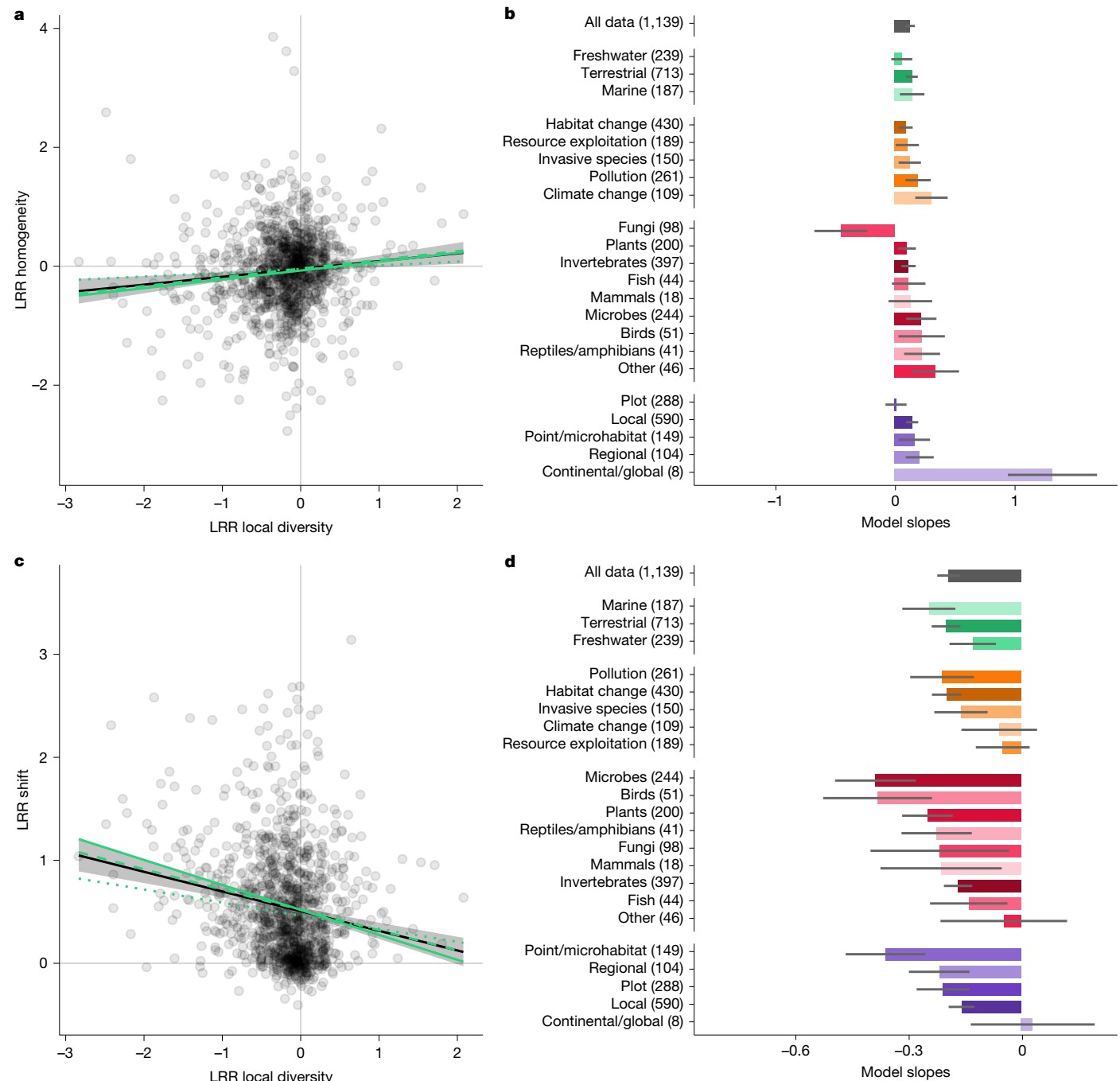

**Fig. 4 | Relationships between local diversity and homogeneity and community composition responses to human pressures. a**, Scatterplot showing the relationship between the log-response ratio (logarithm-transformed ratio of impacted to reference values, LRR) of community local diversity and homogeneity (*n* = 1,139). The black line shows the relation estimated from the linear mixed model and the grey area its 95% confidence interval. The green lines represent the relation estimated for each biome separately (continuous,

marine; dashed, terrestrial; dotted, freshwater). **b**, Model slopes estimated from subsets of each category. Error bar represents standard error. **c**, Scatterplot showing the relationship between the log-response ratio of community local diversity and composition shift (*n* = 1,139). The black line shows the relation estimated from the linear mixed model and the grey area its 95% confidence interval. **d**, Model slopes estimated from subsets of each category. Error bar represents standard error.

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

## Methods

### Search strategy

To detect studies reporting results of homogenization and compositional shift in response to human pressures (systematically comparing impact versus reference scenarios) we conducted an initial bibliographic search on 17 January 2022 on the Science Citation Index Expanded (SCI-EXPANDED) through the Web of Science platform using a combination of 231 relevant keywords (Supplementary Information section 2). The SCI-EXPANDED database was chosen for its functionalities, its generalist aspect and its good coverage of targeted disciplines[49,50]. This initial search generated 73,632 publications and was refined using a two-step procedure (Supplementary Information section 3). First, we searched the full text of all of these publications to detect all mentions of the principal coordinates analysis (PCoA) and non-metric multidimensional scaling (NMDS) methods, which are the main approaches to study and quantify among-community differentiation and composition shift. NMDS and PCoA explicitly aim to represent and preserve the pairwise dissimilarities between samples in the new reduced-dimensional space. Therefore, they were preferred to other ordination methods such as principal component analysis or (detrended) correspondence analysis whose focus is first on summarizing and maximizing the variance or correspondence structure of the data[51]. This step reduced the number of possibly relevant publications to 11,968. Second, we manually and individually evaluated the remaining publications to keep only those that met the minimum inclusion criteria for the meta-analysis. To be included, a publication had to feature at least one plot reporting the projection of biological communities in the first two dimensions of a PCoA or NMDS analysis computed from a community dissimilarity matrix, because both these methods strictly require a distance matrix as input, and are therefore preferred for the graphical representation of beta-diversity. The plot had to report at least two groups, the reference (least impacted) and the impacted groups, which could be visually separated by distinct colours or plotting symbols, respectively. Extracting these two groups from the same plot (coordinate system) allowed us to get quantifiable and comparable data on the magnitude and direction of impact effects directly from the original community level data. Moreover, as this type of plot is much more frequently reported than beta-diversity indices (for example, Sørensen index[13]), this approach makes it possible to include a much larger number and variety of studies. As PCoA and NMDS are directly derived from pairwise beta-diversity matrices, our approach reflects directly the standard beta-diversity indices on which they are built[52], but the extracted information is limited to the first two dimensions of the representation[22]. We excluded restored sites (that is, degraded sites which have been subject to ecological restoration), studies on seed banks and studies with communities reconstructed from stomach content analyses, as they would either not be congruent with the impact framework nor reflect contemporary community change.

The final set of studies included 2,133 publications. A single publication may report one to several (1.72, on average) comparisons between different sets of impacted and reference communities, including comparative and experimental approaches. We extracted several records per publication when results were available for different types of pressure or different taxonomic groups, to systematically include and differentiate several comparisons (for example, effects of different pressures, such as warming and nutrient addition, respectively). We did not create multiple records for cases that were not unambiguously comparable. These cases include: (1) different levels of the same pressure across a gradient (for example, three categories of 'low', 'medium' and 'high' impact for the same pressure). In this case, we extracted the comparison between the two ends of the gradient (for example, low and high), assuming that, in case of experiment/manipulation, authors chose realistic scenarios. (2) Different taxonomic levels (for example, family, genus and species) of the same community. In this case, we took the finest available level (species if possible). (3) Different definitions of the same community (for example, including or excluding non-native species). In this case, we systematically and consistently prioritized the most 'complete' communities (that is, the definition including the largest diversity). When a study reported a comparison only for a subset of a community (for example, without non-native species) we included it only if this subset was consistent between the reference and the impacted sites. (4) Different seasons. In this case, we took only one season (with priority order: spring, summer, autumn, winter). (5) Different time points of a time series. In this case, we systematically took the latest point of the time series.

Each extracted comparison was assigned to one of the three main biomes: terrestrial, marine and freshwater. We grouped human pressures into five main categories (habitat change, pollution, resource exploitation, invasive species and climate change). Organisms were grouped into invertebrates, microbes (including bacteria, archaea and microeukaryotes), plants, fungi, fish, birds, reptiles and amphibians, mammals and others (including complex assemblages of several groups and composite organisms such as lichens). Finally, we classified spatial scales into five categories reflecting the order of magnitude of the geographical extent of the reference sites (that is, the distance between the two furthest reference sites, as the best proxy of how spatially aggregated/dispersed these communities are). These non-overlapping categories have following breaking points: point/microhabitat (sites less than 10 m apart), plot (less than 1 km apart), local (less than 100 km apart), regional (less than 1,000 km apart) and continental/global (more than 1,000 km apart).

Data extraction from the plots was achieved using Webplotdigitizer v.4.6 (ref. 53) in manual mode. If plots had no graduated axes for calibration, we assumed the data to be in an orthonormal space, converting the same number of pixels to the same value for both axes. The number of points extracted per graph ranged from 4 to 664 (mean = 26.7, s.d. = 36.5). In addition to point coordinates extracted from projection plots, we extracted mean local diversity (diversity index) in reference and impacted sites and more information about the study and its design. Further technical information on data collection and extraction is provided in Supplementary Information section 4.

### Effect sizes

We measured the magnitude of homogenization and compositional shift under human pressure, across all the studies. To do so, we used the community coordinates extracted from the projected plots (PCoA and NMDS) to reconstruct a pairwise Euclidean distance matrix between communities from which we computed two effect sizes, following the approach by ref. 22. First, a measurement of the change in community similarity estimated as the log-transformed ratio of the mean distance in impacted communities $\overline{D_I}$ and the mean distance in reference communities $\overline{D_R}$. For clarity, this value is multiplied by $-1$ to express a change in homogeneity (LRR homogeneity) instead of heterogeneity, equation (1). Second, a measurement of the shift in community composition (LRR shift) estimated as the log-transformed ratio of the mean distance between impacted and reference communities $\overline{D_B}$ and the mean distance in impacted and reference communities $\overline{D_W}$, equation (2).

$$\text{LRR homogeneity} = -\ln(\overline{D_I}/\overline{D_R}) \qquad (1)$$

$$\text{LRR shift} = \ln(\overline{D_B}/\overline{D_W}) \qquad (2)$$

Given that these two effect sizes depend on the distances between communities in the analyses and that these distances can be altered by the inclusion of more groups or treatments in the plots, we conducted extra tests based on simulations to ensure the robustness (Supplementary Information section 5). Our results show that LRR homogeneity and LRR shift are generally robust to the inclusion of more groups in the PCoA and NMDS graphs from which they are derived.

For each record, the mean local diversity for the impacted and reference communities was also recorded (if reported as taxonomic richness). We computed a standardized effect size of the change in local

diversity (LRR local diversity) by using the log-transformed ratio of the mean local diversity measured in the impacted communities $\overline{\alpha_I}$ and the mean local diversity in the reference communities $\overline{\alpha_R}$, equation (3).

$$\text{LRR local diversity} = \ln(\overline{\alpha_I}/\overline{\alpha_R}) \tag{3}$$

The three effect sizes described here were not statistically tested individually (that is, within-study statistical testing). Statistical modelling and testing were instead conducted across studies, in which variability was accounted for in the interstudy synthesis, as described in the next section.

### Statistical analyses

We used mixed models to test the effect of biome, pressure, group of organisms and spatial scale on the variation of LRR homogeneity, LRR shift and LRR local diversity. For each effect size, models included all the moderators together as fixed effects and two random effects (intercepts): publications and type of study (field observation, field manipulation or experimental). In all models, publication was used as a random effect to account for between-study heterogeneity. Within-study heterogeneity was not explicitly included because the number of extracted records per study was only 1.72, on average. The models were not weighted to limit the predominance of a few studies on the average effect sizes. As the sample size of LRR homogeneity and LRR shift is the number of pairwise distances between points in each original study and their relationship is exponential, weighting could strongly distort model results. Given the heterogeneity of our dataset, there is a risk that uncontrolled variables, irrespective of the number of datapoints, could bias results if the weighting scheme were to exacerbate their weight. Therefore, we adopted an unweighted approach which in this case is a more robust and more conservative strategy. However, we show that weighting the models using the square root of the sample size yields qualitatively similar results (Extended Data Figs. 8 and 9). Fixed effects were tested using Type II Wald $\chi^2$ tests. All statistical tests reported in this section are two-sided.

Additionally, for each response variable (LRR homogeneity, LRR shift and LRR local diversity), we fitted four independent mixed models including only one moderator at once and the same random effects as described above. From each model we computed estimated marginal means and their 95% and 99% CIs, respectively, for each level of the included factor. Estimated marginal means are an efficient approach to decompose the effect of a specific factor and they were used to produce Figs. 2 and 3. However, comparisons and interpretations of the relative effects of different human pressures must be made with care, as these pressures, which are very different in nature, are exerted on different scales (for example, local chemical pollution versus global warming) and to different degrees that were not measured here.

We used mixed models to test the relationship between LRR local diversity and LRR homogeneity and between LRR local diversity and LRR shift. These two regression models included a random intercept for the original publication. The effects of LRR local diversity on LRR homogeneity and LRR shift were tested using Type II Wald $\chi^2$ tests.

We assessed publication bias by inspecting funnel plots for potential asymmetry driven by small studies[54]. Additionally, we performed a file drawer analysis[55,56] and a P-curve analysis[57] to further confirm the robustness of our findings against selective publication. The results of these analyses (Supplementary Information section 6) indicate that our findings are robust and not unduly influenced by publication bias.

Additionally, we analysed the spatial layout of reference and impacted sites across a subset of studies ($n = 200$), specifically investigating differences between within-treatment distances (spatial homogeneity) and between within and between treatment distances (spatial shift). We found limited but significant evidence of spatial patterns in site distribution. However, these patterns did not correlate with the changes observed in community composition between reference and impacted sites (Supplementary Information section 7).

Statistical analyses were conducted with R v.4.0.3 (ref. 58). Mixed models were fitted using the glmmTMB package[59] and marginal means estimated using the emmeans package[60].

### Reporting summary

Further information on research design is available in the Nature Portfolio Reporting Summary linked to this article.

## Data availability

Articles were searched using the SCI-EXPANDED database accessed through the Web of Science platform (https://www.webofscience.com). Extracted data can be obtained from the GitHub project repository (https://github.com/fkeck/metabeta) and available at Zenodo (https://doi.org/10.5281/zenodo.14608770)[61].

## Code availability

Code to reproduce the analyses and graphical outputs are available on the GitHub project repository (https://github.com/fkeck/metabeta) and available at Zenodo (https://doi.org/10.5281/zenodo.14608770)[61].

49. Gusenbauer, M. & Haddaway, N. R. Which academic search systems are suitable for systematic reviews or meta-analyses? Evaluating retrieval qualities of Google Scholar, PubMed, and 26 other resources. *Res. Synth. Methods* **11**, 181–217 (2020).
50. Foo, Y. Z., O'Dea, R. E., Koricheva, J., Nakagawa, S. & Lagisz, M. A practical guide to question formation, systematic searching and study screening for literature reviews in ecology and evolution. *Methods Ecol. Evol.* **12**, 1705–1720 (2021).
51. Legendre, P. & Legendre, L. F. J. *Numerical Ecology* (Elsevier, 2012).
52. Anderson, M. J. et al. Navigating the multiple meanings of β diversity: a roadmap for the practicing ecologist. *Ecol. Lett.* **14**, 19–28 (2011).
53. Rohatgi, A. Webplotdigitizer v.4.6 (2022).
54. Møller, A. P. & Jennions, M. D. Testing and adjusting for publication bias. *Trends Ecol. Evol.* **16**, 580–586 (2001).
55. Orwin, R. G. A Fail-SafeN for effect size in meta-analysis. *J. Educ. Stat.* **8**, 157–159 (1983).
56. Rosenberg, M. S. The file-drawer problem revisited: a general weighted method for calculating fail-safe numbers in meta-analysis. *Evolution* **59**, 464–468 (2005).
57. Simonsohn, U., Simmons, J. P. & Nelson, L. D. Better P-curves: making P-curve analysis more robust to errors, fraud, and ambitious P-hacking, a reply to Ulrich and Miller. *J. Exp. Psychol. Gen.* **144**, 1146–1152 (2015).
58. R Core Team. *R: A Language and Environment for Statistical Computing* (R Foundation for Statistical Computing, 2022).
59. Brooks, M. E. et al. glmmTMB balances speed and flexibility among packages for zero-inflated generalized linear mixed modeling. *R J.* **9**, 378–400 (2017).
60. Lenth, R. V. Emmeans: estimated marginal means, aka least-squares means. R package version 1.10.0 (2021).
61. Keck F. et al. fkeck/metabeta: v1.0 (v1.0). *Zenodo* https://doi.org/10.5281/zenodo.14608770 (2025).
62. Hedges, L. V., Gurevitch, J. & Curtis, P. S. The meta-analysis of response ratios in experimental ecology. *Ecology* **80**, 1150–1156 (1999).

**Acknowledgements** We thank H. Broadbent for his assistance in digitizing sampling maps. This study was supported by the FAN (Research Talent Development Fund of the UZH Alumni), the Ernst Göhner Foundation, the Gebauer Foundation and the Swiss National Science Foundation (grant no. 310030_197410). L.F. was supported by a stipend (Excellence Stipend for Doctoral Studies) from the Gesellschaft für Forschungsförderung Niederösterreich.

**Author contributions** F.K. designed the study, ran the initial search, performed the initial screening, conducted the analyses and led the writing of the paper. T.P. and F.K. performed the pilot test for minimum inclusion criteria. T.P. and F.A. had a central role in the writing of the paper. F.K., T.P., R.A., C.B., R.B., E.C., T.C., M.C., L.F., D.K., M.K., L.M., R.O., K.T., H.Z. and F.A. participated equally in data extraction and discussed and commented on the manuscript. Apart from F.K., T.P. and F.A., authors are listed in alphabetical order.

**Funding** Open access funding provided by University of Zurich.

**Competing interests** The authors declare no competing interests.

**Additional information**
**Correspondence and requests for materials** should be addressed to François Keck or Florian Altermatt.

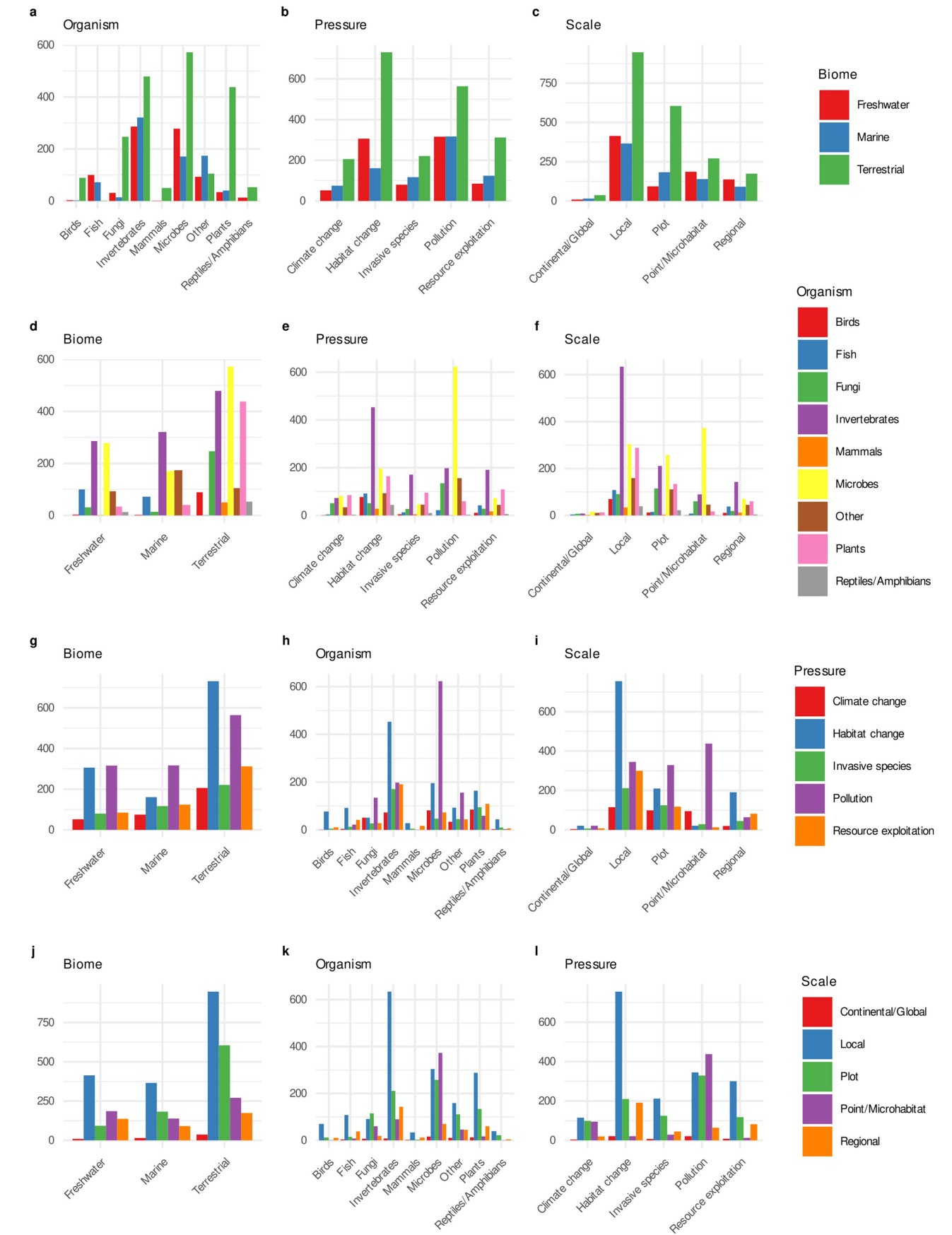

**Extended Data Fig. 1 | Distribution of comparisons of diversity across the four moderators. a–c**, Biomes. **d–f**, Groups of organisms. **g–i**, Human pressures. **j–l**, Spatial scale. For each moderator, the distribution is represented conditionally to the other moderators.

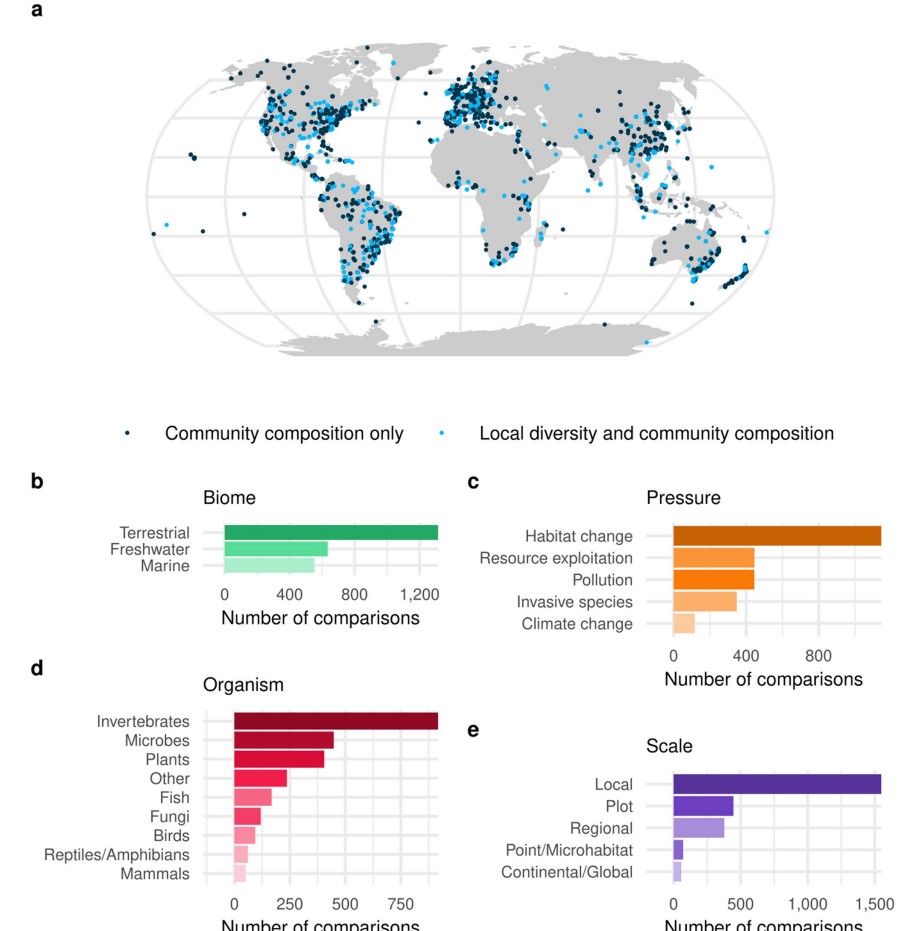

**a**

• Community composition only  • Local diversity and community composition

**b** Biome

Terrestrial
Freshwater
Marine

Number of comparisons

**c** Pressure

Habitat change
Resource exploitation
Pollution
Invasive species
Climate change

Number of comparisons

**d** Organism

Invertebrates
Microbes
Plants
Other
Fish
Fungi
Birds
Reptiles/Amphibians
Mammals

Number of comparisons

**e** Scale

Local
Plot
Regional
Point/Microhabitat
Continental/Global

Number of comparisons

**Extended Data Fig. 2 | Location of diversity comparisons from observational studies and their distribution across biomes, pressures, organisms and scale. a**, Global map of the 2,501 comparisons of diversity included. **b–e**, Distribution of comparisons of diversity by type of biome, human pressures, groups of organisms and spatial scale. These variables correspond to the four main factors tested in this study.

**a**

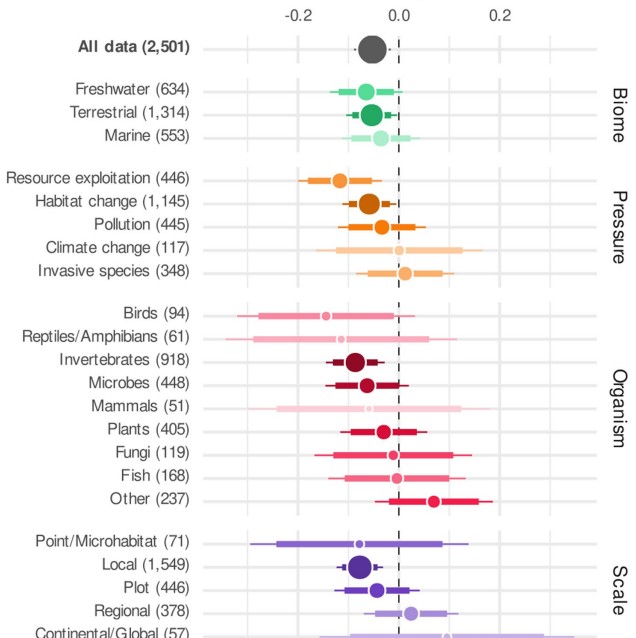

**b**

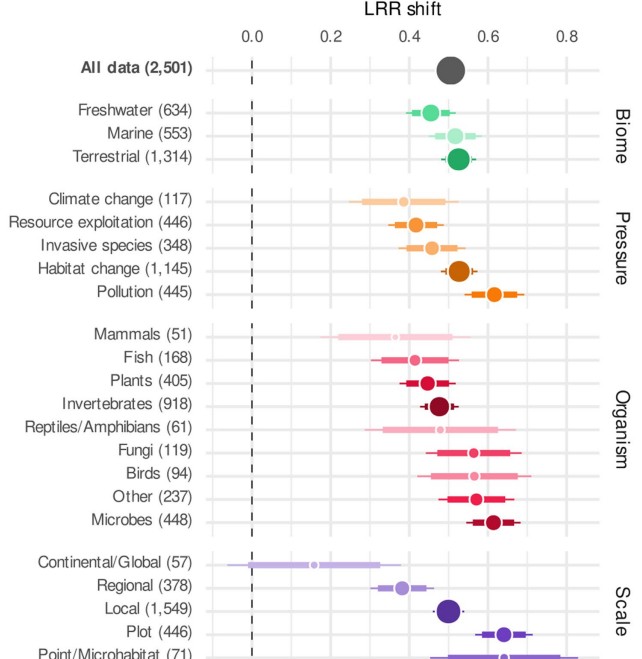

**Extended Data Fig. 3 | Impacts of human pressures on homogeneity and shifts in composition of biological communities estimated from observational studies. a**, ratio of community homogeneity (logarithm-transformed ratio of impacted to reference values, LRR homogeneity). **b**, Log-response ratio of community composition shift (LRR shift). The global response (all data) is shown on the first row of each panel and is separated by factors in the following rows. The numbers between parentheses indicate the number of comparisons. For each category the dot represents the marginal mean computed from the model; dot size is proportional to number of studies included. The larger bar shows the 95% confidence interval and the thinner bar represents the 99% confidence interval.

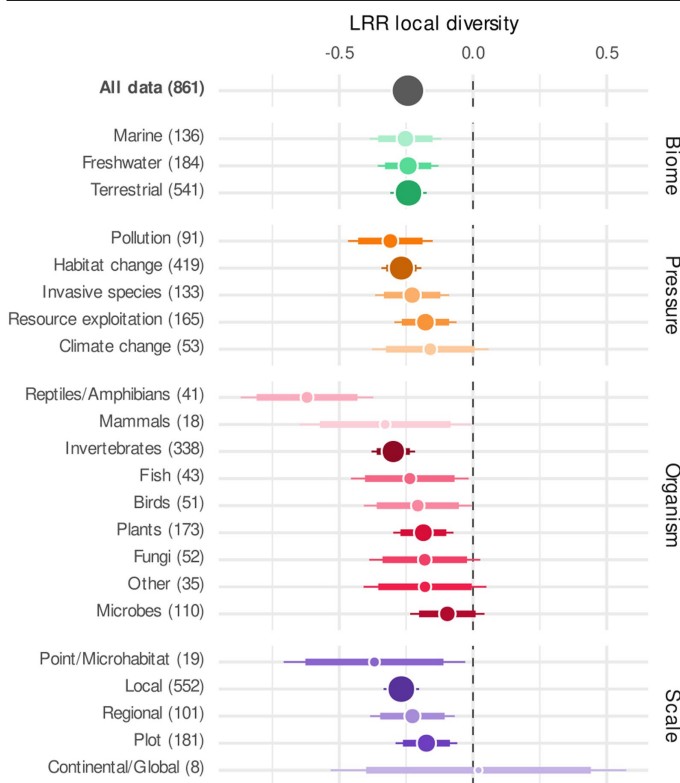

**Extended Data Fig. 4 | Impacts of human pressures on local diversity estimated from observational studies.** Log-response ratio of local diversity (logarithm-transformed ratio of impacted to reference values, LRR local diversity). The global response (all data) shown on the first row is separated by factors in the following rows. The numbers between parentheses indicate the number of comparisons. For each category the dot represents the marginal mean computed from the model; dot size is proportional to number of studies included. The larger bar shows the 95% confidence interval and the thinner bar represents the 99% confidence interval.

**a**

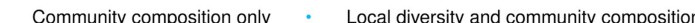

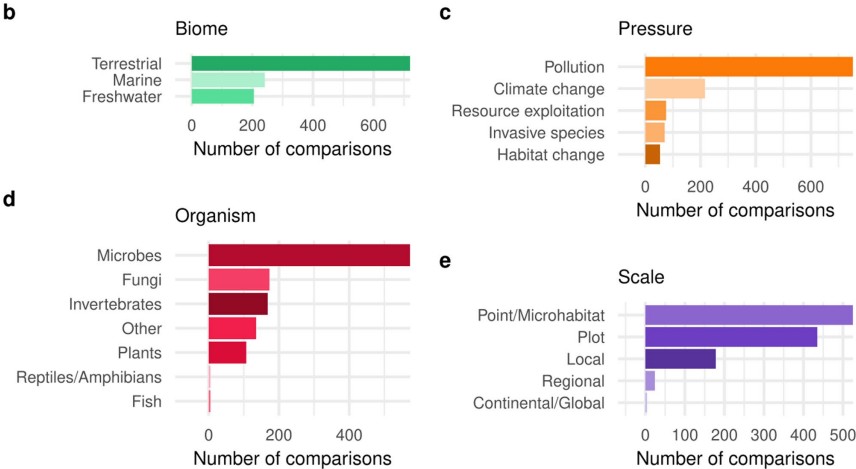

・ Community composition only　　・ Local diversity and community composition

**b**

**Biome**

Terrestrial
Marine
Freshwater

0　200　400　600
Number of comparisons

**c**

**Pressure**

Pollution
Climate change
Resource exploitation
Invasive species
Habitat change

0　200　400　600
Number of comparisons

**d**

**Organism**

Microbes
Fungi
Invertebrates
Other
Plants
Reptiles/Amphibians
Fish

0　200　400
Number of comparisons

**e**

**Scale**

Point/Microhabitat
Plot
Local
Regional
Continental/Global

0　100　200　300　400　500
Number of comparisons

**Extended Data Fig. 5 | Location of diversity comparisons from experimental studies and their distribution across biomes, pressures, organisms and scale. a**, Global map of the 1,166 comparisons of diversity included. **b**–**e**, Distribution of comparisons of diversity by type of biome, human pressures, groups of organisms and spatial scale. These variables correspond to the four main factors tested in this study.

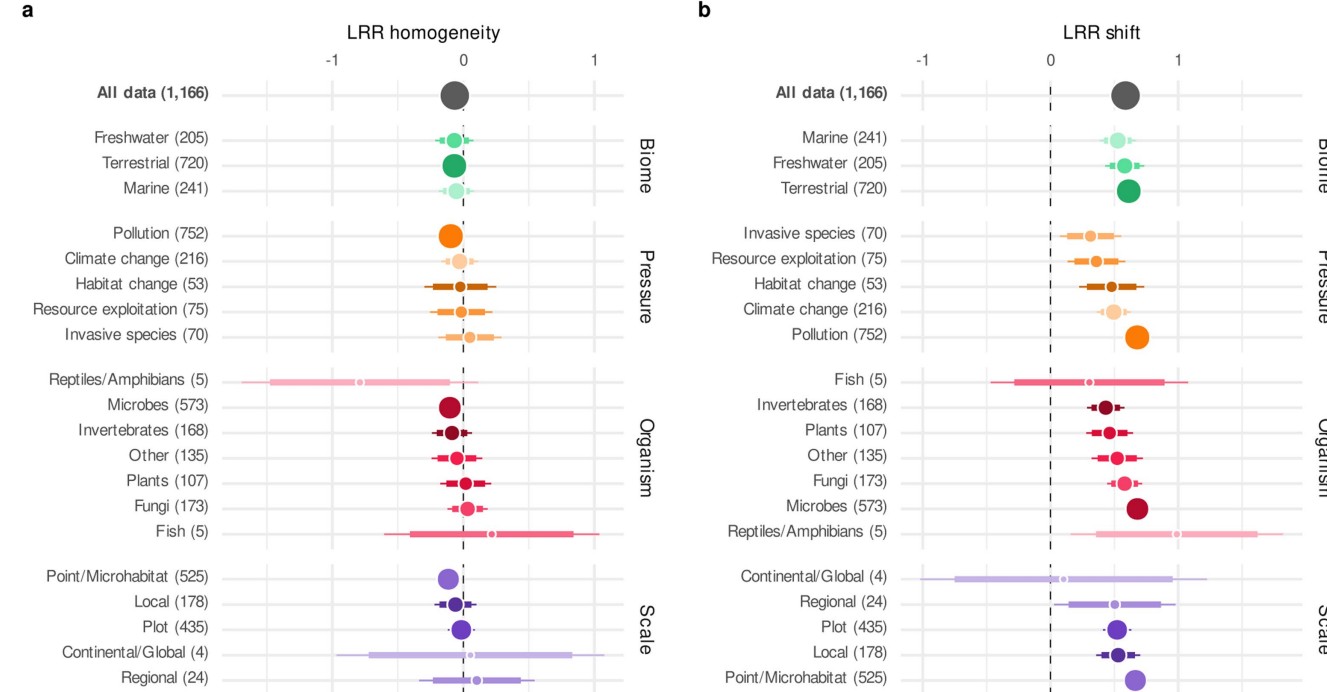

**Extended Data Fig. 6 | Impacts of human pressures on homogeneity and shifts in composition of biological communities estimated from experimental studies. a**, Log-response ratio of community homogeneity (logarithm-transformed ratio of impacted to reference values, LRR homogeneity). **b**, Log-response ratio of community composition shift (LRR shift). The global response (all data) is shown on the first row of each panel and is separated by factors in the following rows. The numbers between parentheses indicate the number of comparisons. For each category the dot represents the marginal mean computed from the model; dot size is proportional to number of studies included. The larger bar shows the 95% confidence interval and the thinner bar represents the 99% confidence interval.

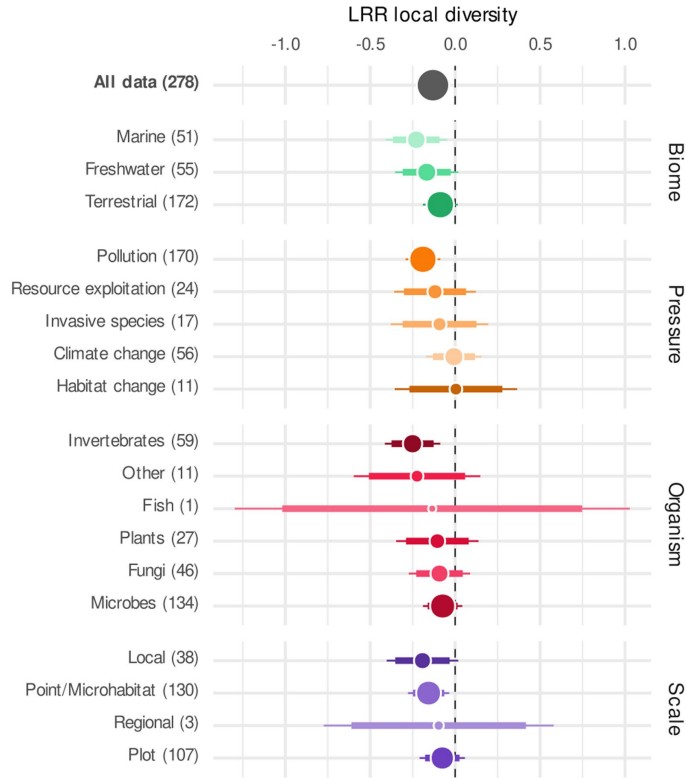

**Extended Data Fig. 7 | Impacts of human pressures on local diversity estimated from experimental studies.** Log-response ratio of local diversity (logarithm-transformed ratio of impacted to reference values, LRR local diversity). The global response (all data) shown on the first row is separated by factors in the following rows. The numbers between parentheses indicate the number of comparisons. For each category the dot represents the marginal mean computed from the model; dot size is proportional to number of studies included. The larger bar shows the 95% confidence interval and the thinner bar represents the 99% confidence interval.

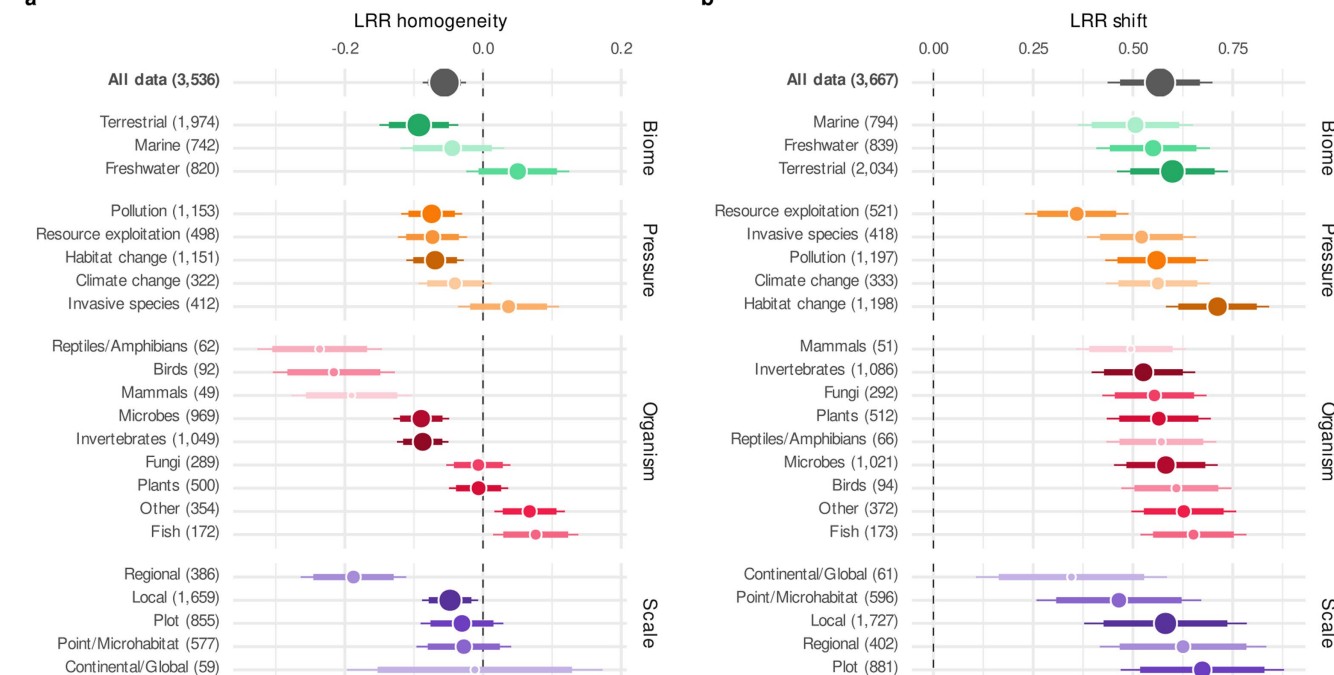

**Extended Data Fig. 8 | Impacts of human pressures on homogeneity and shifts in composition of biological communities estimated using weighted regressions. a**, Log-response ratio of community homogeneity (logarithm-transformed ratio of impacted to reference values, LRR homogeneity). **b**, Log-response ratio of community composition shift (LRR shift). The global response (all data) is shown on the first row of each panel and is separated by factors in the following rows. The numbers between parentheses indicate the number of comparisons. For each category the dot represents the marginal mean computed from the model; dot size is proportional to number of studies included. The larger bar shows the 95% confidence interval and the thinner bar represents the 99% confidence interval. The weighting scheme applied is the inverse of the square root of the variance of the effect sizes, computed following Zhou et al.[22].

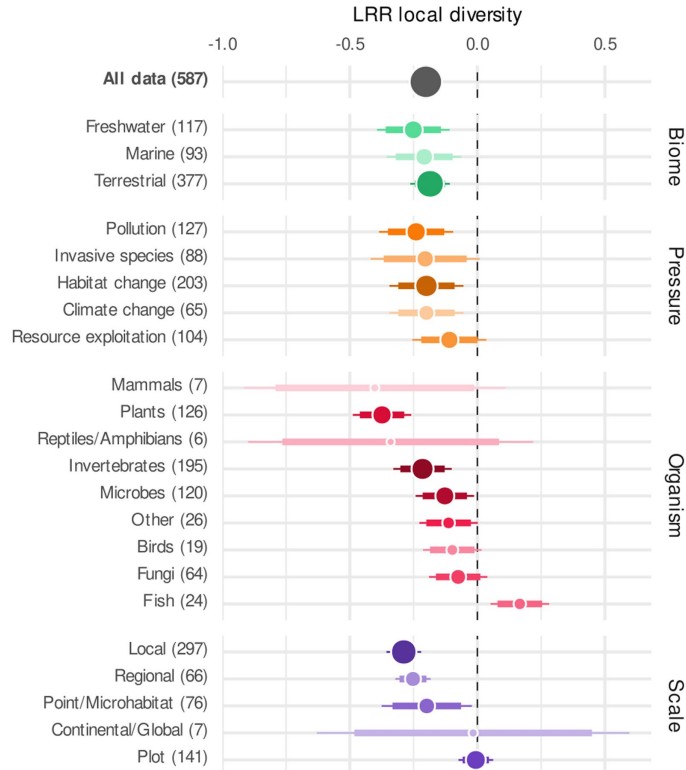

**Extended Data Fig. 9 | Impacts of human pressures on local diversity estimated from experimental studies using weighted regression.**
Log-response ratio of local diversity (logarithm-transformed ratio of impacted to reference values, LRR local diversity). The global response (all data) shown on the first row is separated by factors in the following rows. The numbers between parentheses indicate the number of comparisons. For each category the dot represents the marginal mean computed from the model; dot size is proportional to number of studies included. The larger bar shows the 95% confidence interval and the thinner bar represents the 99% confidence interval. The weighting scheme applied is the inverse of the variance of the effect size, computed following Hedges et al.[62].

Florian Altermatt

# Reporting Summary

## Statistics

For all statistical analyses, confirm that the following items are present in the figure legend, table legend, main text, or Methods section.

| n/a | Confirmed | |
|---|---|---|
| ☐ | ☒ | The exact sample size (*n*) for each experimental group/condition, given as a discrete number and unit of measurement |
| ☐ | ☒ | A statement on whether measurements were taken from distinct samples or whether the same sample was measured repeatedly |
| ☐ | ☒ | The statistical test(s) used AND whether they are one- or two-sided<br>*Only common tests should be described solely by name; describe more complex techniques in the Methods section.* |
| ☐ | ☒ | A description of all covariates tested |
| ☐ | ☒ | A description of any assumptions or corrections, such as tests of normality and adjustment for multiple comparisons |
| ☐ | ☒ | A full description of the statistical parameters including central tendency (e.g. means) or other basic estimates (e.g. regression coefficient) AND variation (e.g. standard deviation) or associated estimates of uncertainty (e.g. confidence intervals) |
| ☐ | ☒ | For null hypothesis testing, the test statistic (e.g. *F*, *t*, *r*) with confidence intervals, effect sizes, degrees of freedom and *P* value noted<br>*Give P values as exact values whenever suitable.* |
| ☒ | ☐ | For Bayesian analysis, information on the choice of priors and Markov chain Monte Carlo settings |
| ☒ | ☐ | For hierarchical and complex designs, identification of the appropriate level for tests and full reporting of outcomes |
| ☒ | ☐ | Estimates of effect sizes (e.g. Cohen's *d*, Pearson's *r*), indicating how they were calculated |

*Our web collection on statistics for biologists contains articles on many of the points above.*

## Software and code

Policy information about availability of computer code

| Data collection | Data extraction from the plots was achieved using Webplotdigitizer v. 4.6 in manual mode. |
|---|---|
| Data analysis | Statistical analyses were conducted with R v.4.0.3. Mixed models were fitted using the glmmTMB package and marginal means estimated using the emmeans package. Code and information about the environment to reproduce the analyses and graphical outputs are available on GitHub (https://github.com/fkeck/metabeta) and are permanently archived on Zenodo (https://doi.org/10.5281/zenodo.14608770). Package version: tidyverse v.2.0.0, readr v.2.1.4, scales v.1.3.0, metafor v.4.4-0, vegan v.2.6-4, patchwork v.1.2.0, MASS v.7.3-55, broom v.1.0.5, car v.3.1-2, broom.mixed v.0.2.9.4, glmmTMB v.1.1.8, emmeans v.1.10.0, ggtext v.0.1.2, jsonlite v.1.8.8, tibble v.3.2.1, grImport2 v.0.3-1, stringr v.1.5.1, glue v.1.7.0, sf v.1.0-15, rphylopic v.1.3.0, ggeffects v.1.5.0, dplyr v.1.1.4, quarto v.1.4 |

For manuscripts utilizing custom algorithms or software that are central to the research but not yet described in published literature, software must be made available to editors and reviewers. We strongly encourage code deposition in a community repository (e.g. GitHub). See the Nature Portfolio guidelines for submitting code & software for further information.

## Data

Policy information about availability of data

All manuscripts must include a data availability statement. This statement should provide the following information, where applicable:
- Accession codes, unique identifiers, or web links for publicly available datasets
- A description of any restrictions on data availability
- For clinical datasets or third party data, please ensure that the statement adheres to our policy

Articles were searched using the Science Citation Index Expanded (SCI-EXPANDED) database accessed through the Web of Science platform (https://www.webofscience.com). Extracted data can be obtained from the GitHub project repository (https://github.com/fkeck/metabeta) permanently archived on Zenodo (https://doi.org/10.5281/zenodo.14608770).

## Research involving human participants, their data, or biological material

Policy information about studies with human participants or human data. See also policy information about sex, gender (identity/presentation), and sexual orientation and race, ethnicity and racism.

| Reporting on sex and gender | Not applicable. |
|---|---|
| Reporting on race, ethnicity, or other socially relevant groupings | Not applicable. |
| Population characteristics | Not applicable. |
| Recruitment | Not applicable. |
| Ethics oversight | Not applicable. |

Note that full information on the approval of the study protocol must also be provided in the manuscript.

# Field-specific reporting

Please select the one below that is the best fit for your research. If you are not sure, read the appropriate sections before making your selection.

☐ Life sciences　　☐ Behavioural & social sciences　　☒ Ecological, evolutionary & environmental sciences

For a reference copy of the document with all sections, see nature.com/documents/nr-reporting-summary-flat.pdf

# Ecological, evolutionary & environmental sciences study design

All studies must disclose on these points even when the disclosure is negative.

| Study description | This study is a meta-analysis of studies comparing biodiversity between reference and human impacted sites. The goal of this study is to measure the magnitude of biodiversity changes under anthropogenic pressure and to understand how these changes relate to different types of human activities. |
|---|---|
| Research sample | All studies reporting results of compositional shift and homogenisation in response to human pressures (systematically comparing impact vs. reference scenarios). |
| Sampling strategy | Our sample size directly depends on available published material. Our approach aims to be exhaustive (i.e. include all available relevant studies, no subsampling). |
| Data collection | The data collection was performed by all co-authors using a web platform developed specifically for the project. Data were collected directly from the publications, using the tool Webplotdigitizer v. 4.6 to extract data from images where necessary. |
| Timing and spatial scale | The initial bibliographic search was performed on the Web of Science database on the 17/01/2022. Data were extracted from publications from June 2022 to February 2023. The publications used in the meta-analysis were published from 1992 to 2022 and cover the entire world (see Fig. 1). |
| Data exclusions | No data were excluded. Only studies that fulfilled the requirements for the meta-analysis were included (see Methods). |
| Reproducibility | All data were extracted from published and available material. Extracted data and the code used for the analyses are available on the GitHub repository of the project: https://github.com/fkeck/metabeta. |
| Randomization | Randomization is not relevant to our study which is a meta-analysis and not an experiment with controlled design. |

| Blinding | Blinding is not relevant to our study which is a meta-analysis. As such, no external participants were involved. |

Did the study involve field work? ☐ Yes ☒ No

# Reporting for specific materials, systems and methods

We require information from authors about some types of materials, experimental systems and methods used in many studies. Here, indicate whether each material, system or method listed is relevant to your study. If you are not sure if a list item applies to your research, read the appropriate section before selecting a response.

## Materials & experimental systems

| n/a | Involved in the study |
|---|---|
| ☒ ☐ | Antibodies |
| ☒ ☐ | Eukaryotic cell lines |
| ☒ ☐ | Palaeontology and archaeology |
| ☒ ☐ | Animals and other organisms |
| ☒ ☐ | Clinical data |
| ☒ ☐ | Dual use research of concern |
| ☒ ☐ | Plants |

## Methods

| n/a | Involved in the study |
|---|---|
| ☒ ☐ | ChIP-seq |
| ☒ ☐ | Flow cytometry |
| ☒ ☐ | MRI-based neuroimaging |

## Plants

| Seed stocks | Not applicable. |

| Novel plant genotypes | Not applicable. |

| Authentication | Not applicable. |

