## [Peer Review File · Nature]

The global human impact on biodiversity

Corresponding Author: Dr Francois Keck

Version 0:

Reviewer comments:

Referee #1

(Remarks to the Author)

This paper carries out a meta-analysis of community responses to human drivers and finds that impacted communities have lower diversity, different composition but are not homogenised. This is a study that could make an important contribution to the growing body of literature on biodiversity change. However I was bit surprised by some important gaps on the literature, some aspects of the methodology, and by some of the claims that seemed at moments to overreach. I detail my comments below.

General comments

1. Contrary to the claims on l.69-79, there has been much work on biodiversity change in impacted sites in relation to controls. PREDICTS has assembled data from almost a thousand studies (cited once in the paper, as confirming the results), but looks mainly at habitat change. A more limited database on impacts of land-use change, but using spatial scaling principles was developed by Chaudhary et al (2015, Env Science Tech). GLOBIO has assessed several other drivers besides habitat loss based on such empirical studies (Alkemade et al 2009, Ecosystems; Schipper et al. 2016 The GLOBIO model. A technical description of version 3.5, PBL Netherlands Environmental Assessment Agency; and references there in). Or for an analysis of plant community similarity across multiple scales and habitats check Staude et al (2021 Ecology Letters), with results contrasting with this study (increasing homogenization). I still find the current analysis goes a step further, by looking at multiple taxa, scales, and drivers simultaneously, but it should be clear this is on the wake of several previous studies.

2. I have three main concerns with the methodology. The first is about how the LRRs homogeneity and shift were calculated. It seems to me that the precision of the estimates for each comparison depends on the number of sites, as they are based on mean distances (within reference sites, within impact sites, and between reference and impact sites). But I did not see an explanation for how many sites are used to calculate each mean distance, if there are minimum number of sites for the study to be included, and if the number of sites is used as a weight.

3. The second is about the use of NDMS and PCA. These are reasonable ways of measuring community shifts and heterogeneity. However they may give different answers from an analysis using Sorensen or other similarity metric. The latter are arguably more common in these kind of studies and at least some discussion of the difference should be added. One problem is that NDMS and PCA depend on the composition of all studies used in the projection. If only a few of the studies from each plot were digitised, the distances may not really reflect the patterns of being studied.

4. Not much is said about the LRR local diversity. How was it measured? As species richness or some other diversity metric? And how were species that were not present in reference sites treated? Some studies exclude "novel" or "invasive" species from such diversity estimates in control-impact studies and this can have major implications for the results.

5. One smaller issue with the methodology is with the issue of spatial scale. The authors use a few different spatial scale categories, but it was not clear to me what happens when the sites straddle multiple spatial scales. For instance in the Breeding Bird Survey one can have pairs of sites closer than 100km, parts of sites between 100km and 1000km, and pairs greater than 1000km apart.

6. Although Control/Impact (CI) studies are useful, the golden standard are Before/After/Control/Impact (BACI) studies. I think a CI study such as this one stands on its own, but would be interesting to add a couple of sentences on potential limitations of not having the before and after measurements at the impact sites.

Specific comments

I.127-128. Several studies are cited about large-scale biotic homogenisation of exotics, but not the one that arguably did the first global analysis of this kind: Capinha et al. 2015 *Science*.

I.128-129. I was left wondering if the LRR of studies looking at the pressure invasive species was calculated with all species or just the exotic species.

I.183-185. I find this speculation a bit baseless. How does one go from shifts in community composition to declines in vertebrate populations?

I.215-217 This seems to be overreaching and it's not clear what kind of benchmark the authors want to provide.

I.188-189. It's not only about null model expectations. It's about the distribution of the sites themselves and the accuracy of the measurements. See Valdez et al. (2023 *Ecography*) study showing limitations on the capacity to detect biodiversity trends from time series.

Referee #2

(Remarks to the Author)

I have been specifically asked to review the methodology of meta-analysis. The authors have not followed the recommended procedures for literature reviews and meta-analyses in ecology and conservation (see e.g. Grames and Elphick 2020 *Biological Conservation* 241: 108385, Foo et al. 2021 *Methods in Ecology and Evolution* 12: 1705-1720, Nakagawa et al. 2022 *Methods in Ecology and Evolution* 13: 4-21; Nakagawa et al. 2023 *Environmental Evidence* 12:8) and as a result multiple details of the review are not clear.

- The authors used Web of Science for literature search and refer to it as a 'database' (line 223). WoS is not a database but a platform which provides access to various reference databases. The specific databases and year spans which are accessible through WoS depend on institutional subscription, hence two researchers from different institutions using the same set of keywords in WoS can end up with different number of hits. To make searches repeatable it is important to report which databases within WoS (e.g. Core Collection) and which years were searched.

- It is normally recommended to conduct searches for meta-analysis in at least two different databases and compare the results. Why WoS only was used and why was it deemed most appropriate for this study (given the focus on human impact some other databases focusing on more applied research could have been used)?

- Who did the searches and study screening checking that the study fits the inclusion criteria? Author contribution statement (lines 425-426) lists the authors involved in data extraction, but it is not clear who did the first two steps on the PRISMA flowchart (publication identification and screening). It is usually recommended that two authors pilot-test the initial screening decision tree and check the degree of agreement between them – has this been done?

- The authors derived their effect sizes from distances measured from PCA and NMDS diagrams. This is not a typical approach to calculating effect sizes in ecological meta-analyses. It is not clear whether the authors have developed this approach by themselves or whether it has been used before in another study. If the former, the justification has to be provided for the new measure of the effect (e.g. it is statistical distribution etc), if the latter, then the references to previous studies using this metric need to be provided. While the extended data Fig 1 is helpful to illustrate how effect size was calculated, I am not clear what is the sample size for this effect – is it the number of pairwise distances? And how the variance for the effect size was computed? Formulas need to be provided.

- Meta-analysis has an established methodology (see e.g. Nakagawa et al. 2023 *Environmental Evidence* 12:8). Effect sizes are normally weighted by the inverse of study variance, it is not clear if any weighting has been done in this analysis and consequently whether different studies had the same or different weight in the analysis. Packages like metafor in R are specifically designed for conducting meta-analyses which allows to implement weighting, fit mixed effects and multilevel models, test for effects of moderators, conduct publication bias tests etc. Why did the authors use instead glmmTNB package which is not specifically suited for meta-analytical models?

- Multiple records were extracted per study (243-244) which might make effect sizes non-independent of each other. Publications were included as a random factor in the models (line 293) presumably to control for this non-independence. A better approach would be to use a multi-level model with study ID and effect ID as random factors which would allow to include both between- and within- study effects as in Nakagawa et al. 2023 *Environmental Evidence* 12:8)

- Why were moderators fitted one at the time (lines 288-289) instead of together (and thus allowing to test for interactions between them)? Has the potential confounding between the moderators been tested?

- Publication bias testing is an important part of the meta-analysis (Nakagawa et al. 2022 *Methods in Ecology and Evolution* 13: 4-21), particularly if it includes only published literature as the given study. No tests for publication bias appear to be conducted.

Referee #3

(Remarks to the Author)

This manuscript describes an ambitious effort to synthesise evidence of how the major direct drivers of biodiversity change affect three important aspects of the composition of ecological communities. They estimate each driver's (a) net effect on

local alpha diversity; (b) effect on compositional change from a reference community, used in lieu of pre-impact communities; and (c) effect on beta diversity, as a way of testing whether the driver causes biotic homogenization or differentiation. They also test whether effects differ among five drivers, nine major taxonomic groups, five different spatial scales, and three 'main biomes' (terrestrial, freshwater and marine). The manuscript is generally very clear, well written and well presented. The logic behind the analysis is laid out very clearly, and Figure 1 is a thing of beauty.

The main results can be summarised as follows. (a) Drivers reduces local diversity significantly, by an average of 13.3% (back-transforming the effect size on line 178), with effect sizes differing among taxonomic groups and drivers. Indeed, pooled effect sizes for resource exploitation and climate change are smaller than those for other drivers and non-significant (Figure 3). (b) All drivers are inferred to cause significant shifts in community composition, in all groups and in all major biomes, with the effects being more pronounced as spatial scale decreases (Figure 3). (c) Effects on community homogeneity are smaller and more complex to summarise: differentiation at smaller spatial scales tends towards homogenization at larger scales, and different taxonomic groups show different tendencies.

There is a lot of novelty here. Although previous syntheses, including some in *Nature*, have shown effects of one or more drivers on local diversity or community composition, I am not aware of any paper that matches the ambition and breadth of this one – with meaningfully large sample sizes for terrestrial, freshwater and marine communities, for each driver, and for such a diversity of taxa. There is an elegance to the analysis, with scale, taxon, driver and 'main biome' all considered simultaneously.

The significance of the paper is less clear. The authors are right not to emphasise the differences in effect sizes among drivers, because dose-response relationships depend on the dose as well as the effect-per-unit-dose, but it does limit the broader impact of the results. I suspect this limitation is why the manuscript is largely framed around the novelty of considering both compositional change and homogenization/differentiation together (lines 73-79), which I did not find a particularly strong motivation.

I also have two fundamental concerns with the data and analysis in the manuscript. One is easy to fix but might change the conclusions relating to all three main results; the other might not be fixable but may simply invalidate results (b) and (c) and all conclusions drawn from them.

First, there is wide recognition that meta-analysis of published literature must consider the possibility of publication bias, i.e., the under-reporting of non-significant effect sizes. The complete omission of any mention of publication bias from this manuscript is therefore unacceptable. It should be tested for using an appropriate method and the results, implications and limitations of the test reported.

Second, and more problematically, I think that a large proportion of the results in (b) and (c) could well be statistical artifacts caused by a ubiquitous ecological phenomenon that the authors mention (lines 134-136) but whose implications for their analysis they do not think through: the 'distance-decay' relationship whereby compositional similarity between two communities falls (monotonically and fairly predictably) with the distance between them, other things being equal.

The authors have gone to admirable and exemplary lengths to capture the compositional distance between the communities at their sites, digitising each site's location in a suitable ordination plot and thence inferring the compositional distance matrix linking all sites. The problem is that geographic distance between sites – which predicts compositional distance so well that it is often used as the starting point for the scaling optimisation – is not considered or controlled for but (crucially) is very likely to differ systematically in ways that bias results (c) and (especially) (b).

Result (b) depends on the untested (and unstated!) assumption that the average geographic distance between an impacted site and a reference site is the same as the average geographic distance between two sites in the same category. This is very unlikely to be met. Studies commonly have multiple sampling points close together within a patch that is treated homogeneously. Sometimes this is pseudoreplication that goes entirely unrecognised through analysis and peer review (the effective sample size might really be one comparison between a block of impacted sites and a block of reference sites), or sometimes there is a higher-level structure to the design (with the pseudoreplicated samples being pooled prior to testing hypotheses). Whenever this happens, the average within-condition distance will be less – often much less – than the average between condition distance. (Conversely, if the design has spatial blocks each with only one reference and one impacted site, the bias will be the other way; my experience is that such designs are rarer however.)

Result (c) depends just as strongly on the untested assumption that the average geographic distance between two impacted sites is the same as the average geographic distance between two reference sites. This assumption will also be violated under many commonly used study designs. A common design in heavily-impacted settings is that the same patch of reference habitat is used to make multiple reference-impact comparisons, using paired comparisons of sites within each of which a different site just inside the reference patch is compared with a nearby site just outside. In such a design, the mean reference-reference geographic distance is smaller than the mean impact-impact difference, biasing the inference towards differentiation. Likely circumstances for this kind of design would be studies assessing effects of protected areas (where resource exploitation should be lower) on community composition.

I see two possible ways forward with this problem. First, it may be possible, for at least a subset of studies, to use maps or tables of geographic coordinates to compare the three kinds of geographic distances. My objection obviously falls if there is no bias or if the bias opposes rather than drives the results, and I would be supportive of publication in *Nature*. Second, the issue of distance-decay needs to be foregrounded and the resulting limitations given prominence. There would still be

enough novelty, ambition and quality for the resulting manuscript to be published in an excellent journal in that case, but I don't think it would be competitive for Nature.

I also have a few very much less fundamental suggestions that the authors may find helpful for improving the manuscript. In manuscript order, these are:

Lines 67-76 imply (perhaps unwittingly) that the data are all time series from matched sets of reference and impacted sites but, as lines 254-255 make clear, time series were not used (in that only the last year of them was used). Please clarify.

Lines 234-237: I briefly thought that this sentence included an absolutely fatal flaw but now think that's a misreading of an ambiguous statement. My (hopefully) mistaken reading was that you discarded sources where the reference and impacted groups of sites were not different ("could not be visually separated"), but I assume that you meant simply that the plot had to differentiate sites from the two conditions by colour or plotting symbol. But please clarify so no-one else leaps to the same wrong conclusion.

Lines 269-271: A range of different diversity measures were used. Although a random intercept was fitted to accommodate heterogeneity arising from this range, did you use any one particular 'version' of a diversity index in preference to others? (e.g. consider the range of ways Simpson's index gets presented.)

Referee #4

(Remarks to the Author)

This paper presents a meta-analysis of global studies on the biodiversity impacts of five important aspects of human action: habitat alteration, resource exploitation, pollution, climate change and invasive alien species. It brings together data from a wide range of experimental and observational studies, but restricts itself to those that explicitly compare impacted and (relatively) pristine sites, and specifically to those that consider community change using one of two ordination methods. The result is the sort of broad-spectrum, general interest human impact assessment that is likely to garner wide interest.

I am not a specialist on the particular methodology employed, but it seemed largely sensible. I would advise enlisting a review from someone steeped in this approach (I will make recommendations in comments to the Editor). I was, however, curious about the choice of ordination methods to include here; while PCoA and NMDS are widely used, many other methods, such as DCA (or DECORANA), are also widely used in the literature; little justification is given for why the two focal measures are the only ones appropriate here. Also, despite frequent mentions of assessing "Local diversity", I was unable to determine whether the authors meant raw species richness, some diversity index (Shannon? Simpson?) or something else (Chao index? Rarified richness?). The Methods section (lines 269-270) indicates that some sort of "diversity index" is used, but perhaps that means that different studies were assessed using different measures (if so, how were these differences reconciled?) It was also not clarified whether biodiversity impacts of invasive species counted the alien species themselves in the diversity measures.

The paper's findings were interesting, and mostly in line with the scientific consensus. It is certainly not surprising to find that the various human interventions significantly shift community composition, and that most have adverse effects on local diversity (whatever that means here). The authors suggest (lines 153-154) that the community changes seen are in "a consistent direction" -- but no detail is given as to the nature of that directional change, nor whether it is consistent across drivers. The most striking finding, however, was a lack of impact on biotic homogeneity (here operationally defined as scatter on an ordination plot). It was also interesting (but little mentioned in the text) that climate change impacts on diversity appeared quite weak, but of course it is difficult to find appropriate "control" sites for comparisons in this case.

These latter points raised a question in my mind. The dataset includes a large fraction (roughly 1/3) of experimental studies (perhaps a higher fraction for climate change?). This in itself is a strength, but it also raises some concerns of possible experimental artefacts. It has long been known, for instance, that experimental studies on plant species invasions provide contrasting results when compared to observational studies. In part this may be because experiments tend to be performed at much finer spatial scales than many observational studies (although spatial grain is one of the issues analysed here). However, a second concern is that experimental plots are frequently isolated from one another by dissimilar habitat, a feature which might increase beta diversity between plots and thus reduce biotic homogeneity. It would be helpful if the authors could differentiate experimental and observational studies in their analysis (at least in the extended data section).

Overall, I found this an interesting paper, and one making a potentially useful contribution. Nonetheless, it requires a bit more methodological clarity and circumspection before publication in a journal of this calibre could be warranted.

Version 1:

Reviewer comments:

Referee #1

(Remarks to the Author)

I would like to congratulate the authors for the tremendous work they did in replying to my comments on the previous draft of the manuscript. It was truly a pleasure to read the reply. I only have one very minor comment below:

(1) You state "As PCoA and NMDS are directly derived from pairwise beta-diversity matrices, " (l.254-259). Is this always the case, this is, you only use PCoA and NDMS from pairwise distance matrices or in some cases the PCoA where done directly on the original community matrices?

I look forward to see this paper in print.

(Remarks on code availability)

Referee #2

(Remarks to the Author)

I appreciate a very thorough revision of the ms made by the authors. All my comments have been addressed, in particular new sensitivity analyses have been conducted to test how results from unweighted models differed from those of the weighted models and publication bias tests have been conducted. Other methodological details concerning the meta-analysis have been clarified as well. I think the ms has greatly improved as a result of revision.

(Remarks on code availability)

Referee #4

(Remarks to the Author)

I was positively impressed with this manuscript when I first reviewed it, and the revised manuscript is substantially improved, with clearer and more informative prose (especially as to methodologies). All of my original substantive concerns have been addressed adequately.

A few additional points:

(a) The order in which the different sets of results are presented shifts over the course of the paper. In my mind, the "logical" order would be first diversity ("D": how many types), then composition ("C": WHICH types), then homogenisation ("H": similarity between samples in which types), but that is seldom used here. Instead it is sometimes CHD (lines 42-43), sometimes CDH (lines 44-46), sometimes DCH (lines 55-59), and then ultimately HCD (lines 110-114 and thereafter). In the methodology section (lines 308-326) it shifts again: back to CHD. This isn't important, but it is sometimes a bit confusing.

(b) The presentation of the LRR homogeneity results seems peculiar, with verbal interpretations given for very NON-significant trends (e.g. for effects of Invasive spp, line 135), whereas the strongly significant (de-homogenising) effects of drivers such as resource exploitation are not mentioned. The fact that LRR homogenisation interacts with spatial scale is particularly interesting, but the text (lines 128-130) gives equal emphasis to the consistent and significant de-homogenising patterns found at fine spatial scales and the inconsistent and non-significant homogenising patterns at coarse scales. It also doesn't mention that these scaling properties are quite different for the experimental versus observational studies, a fact that might shed light on the discussion of stochasticity and drift (lines 140-142), as well as on the inference of causality (lines 212-213).

(c) I am left slightly mystified by the interpretation of the "LRR shift" results. If I understand correctly, these refer to log response ratios in community compositional differences ("shifts") between impacted and reference plots, relative those found within the sets of impacted and reference communities (see lines 308-313). This comparison of variance between groups to variance within groups is familiar (e.g. in ANOVA), but it includes no direct measure of community changes over time relative to some baseline (community "shifts" per se), as to do so the data would need to include prior community compositions (as in a BACI design, which the paper makes clear in lines 197-202 is NOT used here). Thus so long as the treatment groups differ from each other, the "shift" will be attributed to the "impacted" group rather than the reference group. I am also somewhat confused about the appropriate degree of statistical confidence to claim in analyses of this sort. A set of N values will have $(N^2 - N)/2$ unique within-group non-self comparisons, and those between treatments will have $(N_1 \times N_2)/2$ unique comparisons. These comparisons are not fully independent, as e.g. a single highly unusual community creates a large number of long distance values. Surely there are specialised methods (e.g. Mantel tests) for distance matrices of this sort?

Overall, however, I find the paper much improved, and expect it will make a strong contribution to the growing literature on human biodiversity impacts. I am happy to recommend its publication, once any remaining (mostly minor) issues are addressed.

(Remarks on code availability)

Version 2:

Reviewer comments:

Referee #4

(Remarks to the Author)

I have twice reviewed this manuscript positively, and after each round of reviews, the authors have made substantial revisions, further improving the paper. I am happy for the manuscript to proceed to publication, subject only to minor editorial

corrections where needed.

This is an interesting and ambitious paper, and I have no doubt that it will spark useful discussions.

(Remarks on code availability)

Authors' Response to Reviews of The global human impact on biodiversity

F. Keck *et al.*

Submitted to *Nature* [2023-11-21089]

Referee #1

This paper carries out a meta-analysis of community responses to human drivers and finds that impacted communities have lower diversity, different composition but are not homogenised. This is a study that could make an important contribution to the growing body of literature on biodiversity change. However I was bit surprised by some important gaps on the literature, some aspects of the methodology, and by some of the claims that seemed at moments to overreach. I detail my comments below.

Thank you very much for your review. We sincerely appreciate your positive feedback and your insightful comments. We have carefully considered your recommendations and we have revised the manuscript accordingly.

Specifically, we have ensured that the literature is more broadly covered, refined and clarified the methodology, and clarified how all our statements are appropriately and quantitatively supported by the data and our analysis to have appropriate and corroborated conclusions (see our replies to your comments below where we further detail these changes). We have also ensured that our statements are not perceived as overreaching by clarifying how our claims are based on factual findings. These revisions have strengthened the overall quality of our study and we are grateful for your constructive contribution to this process.

General comments

1.1. Contrary to the claims on l.69-79, there has been much work on biodiversity change in impacted sites in relation to controls. PREDICTS has assembled data from almost a thousand studies (cited once in the paper, as confirming the results), but looks mainly at habitat change. A more limited database on impacts of land-use change, but using spatial scaling principles was developed by Chaudhary et al (2015, Env Science Tech). GLOBIO has assessed several other drivers besides habitat loss based on such empirical studies (Alkemade et al 2009, Ecosystems; Schipper et al. 2016 The GLOBIO model. A technical description of version 3.5, PBL Netherlands Environmental Assessment Agency; and references there in). Or for an analysis of plant community similarity across multiple scales and habitats check Staude et al (2021 Ecology Letters), with results contrasting with this study (increasing homogenization). I still find the current analysis goes a step further, by looking at multiple taxa, scales, and drivers simultaneously, but it should be clear this is on the wake of several previous studies.

> You are absolutely right that our study is based on previous works, and indeed there is extensive work on biodiversity change, for which we have now added your suggested further references (Alkemade et al. 2009, Newbold et al. 2015, Chaudhary et al. 2015, Schipper et al. 2020, Staude et al. 2022). Given that the field of biodiversity change is such a large field, it is always a matter of personal preference on how broad or more focused to start opening paragraphs. We are happy to present our manuscript in a broader context, also acknowledging the broader readership of Nature.

To accomplish this, we have specifically reformulated the paragraph you highlight to take account of your comment and to include the references you suggested. We now use the introduction to give due credit to a selection of highly important and relevant publications related to the topic, adding all the papers you suggested and further building on your suggested papers (if there are other papers that you find fit better, we are happy to consider them). As you state, our analyses go an important step further than previous work, looking at multiple taxa, scales and drivers simultaneously. We have now ensured this novelty of our study is clear and embedded in the previous body of work.

Manuscript L.69–77

Crucially, previous research attempting to generalise biodiversity change have hitherto neglected two key elements. Firstly, most past studies looked at biodiversity change across time using individual time-series and did not contrast findings to reference controls⁹. Secondly, previous studies have rarely differentiated between changes in local diversity versus changes in variation in diversity across space. Unfortunately, the studies that have integrated these elements are generally restricted to a certain type of pressure or to a particular biome^{15–19}. Consequently, we lack generalisations on the effects of human pressures on ecosystems and our understanding of biodiversity change with regard to its different dimensions remains incomplete.

Alkemade, R. et al. GLOBIO3: A Framework to Investigate Options for Reducing Global Terrestrial Biodiversity Loss. *Ecosystems* 12, 374–390 (2009).

Chaudhary, A., Verones, F., de Baan, L. & Hellweg, S. Quantifying Land Use Impacts on Biodiversity: Combining Species–Area Models and Vulnerability Indicators. *Environ. Sci. Technol.* 49, 9987–9995 (2015).

Newbold, T. et al. Global effects of land use on local terrestrial biodiversity. *Nature* 520, 45–50 (2015).

Schipper, A. M. et al. Projecting terrestrial biodiversity intactness with GLOBIO 4. *Global Change Biology* 26, 760–771 (2020).

Schipper, A. M. et al. Projecting terrestrial biodiversity intactness with GLOBIO 4. *Glob. Change Biol.* 26, 760–771 (2020).

Staude, I. R. et al. Directional turnover towards larger-ranged plants over time and across habitats. *Ecology Letters* 25, 466–482 (2022).

1.2. I have three main concerns with the methodology. The first is about how the LRRs homogeneity and shift were calculated. It seems to me that the precision of the estimates for each comparison depends on the number of sites, as they are based on mean distances (within reference sites, within impact sites, and between reference and impact sites). But I did not see an explanation for how many sites are used to calculate each mean distance, if

there are minimum number of sites for the study to be included, and if the number of sites is used as a weight.

> This is a very good comment, and we agree that sample size can impact the precision of the estimates. However, there are two specific reasons why we did not weight our analyses by sample size.

Firstly, as noted by reviewer 2, the sample size of our effect sizes is the number of pairwise distances between points within each original study (which is very different to classic meta-analyses that build generally on mean and SD/SE estimates, that is, these have one value per study). This number of pairwise distances per actual data points increases exponentially with the number of points in the plots. In our case, this means that one study can weigh 10–50,000 times more than another (e.g., a study that has 200 points and thus 39,800 pairwise extracted estimates, compared to a study with only a few data points and thus few pairwise estimates). Consequently the weighting would strongly—and not justifiably—distort model results. We have now added a sentence to the manuscript indicating that the extraction of the pairwise comparisons (instead of overall means as classically done) does not justify a correction on effect sizes based on sample size.

Secondly, the very broad scope of our meta-analysis makes the data set very heterogeneous. Therefore, there is a risk that uncontrolled variables, irrespective of the number of datapoints/pairwise comparisons, could bias results if the weighting scheme were to exacerbate their weight. Also because of this high heterogeneity, less replicated local studies made under experimental conditions can be given a lower weight than large scale field studies with large sample sizes, although the accuracy of estimates for the former is probably better than for the latter. We also added this information to the supplementary material.

Manuscript L.341–350

The models were not weighted to limit the predominance of a few studies on the average effect sizes. As the sample size of LRR homogeneity and LRR shift is the number of pairwise distances between points within each original study and their relationship is exponential, weighting could strongly distort model results. Given the heterogeneity of our data set, there is a risk that uncontrolled variables, irrespective of the number of datapoints, could bias results if the weighting scheme were to exacerbate their weight. Therefore, we adopted an unweighted approach which in this case is a more robust and more conservative strategy. However we show that weighting the models using the square root of the sample size yields qualitatively similar results (Supplementary Information section S5).

For these two reasons, adopting an unweighted approach is a more robust and more conservative strategy, as it is less subject to factors beyond our control. This choice is also in line with recent research more generally questioning the weighting of effect sizes in meta-analyses (see e.g. Buck et al. 2022 MEE, Hamman et al. 2018 Ecosphere).

Despite all these reasons for not weighting, we still re-did the analyses to see any effect weighting would have on the general conclusions of our manuscript (see Figure below, where we used the square root of the sample size for a more realistic weighting scheme). As shown in the figure below (which gives the same analysis as in the main manuscript, yet with a weighting included), the result is qualitatively all the same, that is, even when correcting/considering for sample size, we come to the same conclusions. We have now added these additional analyses to the supplement.

Figure S10 Impacts of human pressures on homogeneity and shifts in composition of biological communities estimated using weighted regressions. The weighting scheme applied is the inverse of the square root of the variance of the effect sizes, computed following Zhou et al.1 . a, Log-response ratio of community homogeneity (logarithm-transformed ratio of impacted to reference values, LRR homogeneity). b, Log-response ratio of community composition shift (LRR shift). The global response (all data) is shown on the first row of each panel and is separated by factors in the following rows. The numbers between parentheses indicate the number of comparisons. For each category the dot represents the marginal mean computed from the model; dot size is proportional to number of studies included. The larger bar shows the 95 % confidence interval and the thinner bar represents the 99 % confidence interval.

Figure S11 Impacts of human pressures on local diversity estimated from experimental studies using weighted regression. The weighting scheme applied is inverse of the variance of the effect size, computed following Hedges et al.2 . Log-response ratio of local diversity (logarithm- transformed ratio of impacted to reference values, LRR local diversity). The global response (all data) shown on the first row is separated by factors in the following rows. The numbers between parentheses indicate the number of comparisons. For each category the dot represents the marginal mean computed from the model; dot size is proportional to number of studies included. The larger bar shows the 95 % confidence interval and the thinner bar represents the 99 % confidence interval.

Finally, you asked if there was a minimal number of study points per study required for the study to be included. Given that any such “minimal number threshold” would be partially arbitrary, we had decided against excluding any studies based on their sample size, as we think every bit of information is worth including. On average, we extracted 26.7 (sd=36.5) points per plot. Based on your comment and reviewer 2's comments we have clarified our methodological choices in the Methods section.

Manuscript L.298

The number of points extracted per graph ranged from 4 to 664 (mean = 26.7, sd = 36.5).

Given that the additional analyses above on correcting for sample size (square root on number of comparisons/study) give qualitatively the same results, this also strongly indicates that the studies with lower numbers of comparisons are not qualitatively different from the other studies, and that the overall patterns we found are robust and general.

Hamman, E. A., Pappalardo, P., Bence, J. R., Peacor, S. D. & Osenberg, C. W. Bias in meta-analyses using Hedges' d. *Ecosphere* 9, e02419 (2018).

Buck, R. J., Fieberg, J. & Larkin, D. J. The use of weighted averages of Hedges' d in meta-analysis: Is it worth it? *Methods in Ecology and Evolution* 13, 1093–1105 (2022).

1.3. The second is about the use of NDMS and PCA. These are reasonable ways of measuring community shifts and heterogeneity. However they may give different answers from an analysis using Sorensen or other similarity metric. The latter are arguably more common in these kind of studies and at least some discussion of the difference should be added. One problem is that NDMS and PCA depend on the composition of all studies used in the projection. If only a few of the studies from each plot were digitised, the distances may not really reflect the patterns of being studied.

> Firstly, on your comment regarding the Sorensen index and other similarity metrics: we deliberately made the choice to use dimension reduction methods because too few studies report global beta-diversity metrics such as Sorensen's index (Petsch et al. 2022 *Oikos*, for example, could find only 43 studies using beta-diversity metrics to investigate homogenisation driven by invasive species). Deriving this information from NMDS and PCoA projection plots therefore enables us to greatly increase the number of included studies and to investigate several co-factors in detail. To address your comment, we have added two sentences in the Methods section to clarify the logic behind our approach and to indicate that the results may differ slightly from traditional beta-diversity indices.

Manuscript L.254–259

Moreover, as this type of plot is much more frequently reported than beta-diversity indices (e.g. Sørensen index¹³), this approach makes it possible to include a much larger number and variety of studies. As PCoA and NMDS are directly derived from pairwise beta-diversity matrices, our approach reflects directly the standard beta-diversity indices on which they are built⁵² but the extracted information is limited to the first two dimensions of the representation²².

Secondly, regarding your comment on extracting specific groups from multivariate analyses: as explained in the Methods section, the chosen approach specifically focussed on extracting the two groups of samples that could be directly and meaningfully compared, namely reference and impact treatment. Any further points or groups of points, such as intermediate or non-contrast treatments were not informative in this sense, and thus omitted. We have made the assumption that the presence of other groups of points in the plots does not pose a problem for the estimation of the effect sizes that we use. In fact, if the addition of communities to a PCoA/NMDS does indeed modify the estimated configuration and the absolute distances between points, we can assume (because this is one of the objectives of these analyses) that the relative distances are correctly conserved. We appreciate your comment and to address it, we have: 1) added a simulation proving the robustness of the effect sizes in this situation and 2) modified the wording of a few sentences in the methods to ensure our extraction approach is clear. We give now the below text/figures in the Supplementary Information S4.

Manuscript L.317–322

Given that these two effect sizes depend on the distances between communities in the analyses and that these distances can be altered by the inclusion of additional groups or treatments in the plots, we conducted additional tests based on simulations to ensure the robustness (Supplementary Information section S4). Our results show that LRR shift and LRR homogeneity are generally robust to the inclusion of additional groups in the PCoA and NMDS graphs from which they are derived.

Supplementary Information section S4

Our meta-analysis is based on the extraction of sample coordinates from PCoA or NMDS two-dimensional graphs. It is common for several groups of samples or treatments to be represented on the same graph, which led us either to extract several comparisons per graph, or to ignore certain groups of samples (the rules applied for this choice are explained in the “Search strategy” part of the Methods section of the manuscript). The inclusion of additional samples or groups of samples can modify the configuration of the points as calculated by the projection methods, with the possible consequence that the observed distances no longer truly reflect the patterns studied. The impact of other sample groups in the analyses is a priori limited since our effect sizes (LRR homogeneity and LRR shift) depend on the relative distances and not the absolute distances between points. We show by simulation that this effect is indeed limited. Our approach is as follows. We randomly generate a set of communities distributed in n groups ($3 \leq n \leq 5$), each group being defined by its own environmental conditions. We then compute first PCoA (or NMDS) for the communities belonging to the first two groups, from which we calculate the LRR homogeneity and LRR shift effect sizes. We then compute a second PCoA (or NMDS), this time with all the groups, from which we calculate the LRR homogeneity and LRR shift effect sizes for the first two groups only. The comparison of LRR homogeneity and LRR shift between the two PCoAs allows us to estimate the bias induced by the presence of other groups in the analysis. Details of the community simulation are given in the following pseudo-code (the R code used is available on GitHub).

Listing S1 Algorithm for community simulation

1. Determine the number of groups, n , an integer randomly chosen in [3, 4, 5].
2. Determine the number of communities for each group, where each value is an integer randomly chosen in [20, ..., 200]

3. Determine the number of environmental dimensions, m , an integer randomly chosen in [3, 4, 5].
4. Determine the number of species in the pool, s , an integer randomly chosen in [15, ..., 100], and create S an indexed collection of species of size s .
5. Create a grouping vector, G , where each element G_i is an integer in [1, ..., n] and represents the group membership of each communities C_i .
6. Generate an $s \times m$ random matrix M with each element independently sampled from a uniform distribution in the range [-1, 1], representing the means of the normal distributions of each species along each environmental dimension.
7. Generate an $s \times m$ random matrix V with each element independently sampled from a uniform distribution in the range [0, 1], representing the standard deviation of the normal distributions of each species along each environmental dimension.
8. Generate an $n \times m$ random matrix E containing environmental conditions for each group along each environmental dimension, with each element independently sampled from a normal distribution $\mathcal{N}(\mu = 0, \sigma^2 = 0.05)$.
9. For each community C_i belonging to the group G_i , generate species abundances by randomly drawing 300 individuals from S , where for each species S_j the probability to be sampled is weighted by the product of the densities of its normal distributions along each environmental dimension $\forall k \in \{1, \dots, m\}$, $\mathcal{N}(\mu_{j,k} = M_{j,k}, \sigma_{j,k}^2 = V_{j,k}^2)$ corresponding to the environmental conditions of the community $E_{G_i,k}$.

In total, we simulated 1000 independent sets of communities. Overall, we found a strong correlation between LRR homogeneity derived from the 2 groups analysis and LRR homogeneity derived from the >2 groups analysis (Figure S8a,c). This correlation was $\rho = 0.85$ ($t = 51.53$, $df = 998$, $P < 0.001$) for PCoA and $\rho = 0.70$ ($t = 31.176$, $df = 998$, $P < 0.001$) for NMDS. Figures S9a and S9c show the distribution of the difference between the two variables for the PCoA and NMDS analyses. The mean of this difference is 0.004 for PCoA and -0.007 for NMDS. One sample t-tests show that neither value is statistically different from zero (PCoA: $t = 0.984$, $df = 999$, $P = 0.325$; NMDS: $t = -1.371$, $df = 999$, $P = 0.170$), indicating the absence of bias.

We also found a very strong correlation between the LRR shift values derived from the 2 groups and the >2 groups analyses (Figure S8b,d). This correlation was $\rho = 0.99$ ($t = 218.27$, $df = 998$, $P < 0.001$) for PCoA and $\rho = 0.83$ ($t = 31.176$, $df = 998$, $P < 0.001$) for NMDS. Figures S9b and S9d show the distribution of the difference between the two variables for the PCoA and NMDS analyses. The mean of this difference is -0.035 for PCoA and 0.063 for NMDS. One sample t-tests show that both values are statistically different from zero (PCoA: $t = -9.419$, $df = 999$, $P < 0.001$; NMDS: $t = 3.447$, $df = 999$,

$P < 0.001$), indicating a very small but consistent bias, possibly driven in the case of the NMDS by the series of outliers observed in Figure S8d and S9d.

Figure S8 Relationship between the effect-sizes estimated from an analysis based on two groups of points and the same effect-sizes estimated from the same analysis but including additional groups of points. a. LRR homogeneity from PCoA, b. LRR shift from PCoA, c. LRR homogeneity from NMDS and d. LRR shift from NMDS. The black line delineate the 1:1 line, corresponding to perfect correspondence between the two estimates.

Figure S9 Distributions of the differences between the effect-sizes estimated from an analysis based on two groups of points and the same effect-sizes estimated from the same analysis but including additional groups of points. **a.** LRR homogeneity from PCoA, **b.** LRR shift from PCoA, **c.** LRR homogeneity from NMDS and **d.** LRR shift from NMDS.

1.4. Not much is said about the LRR local diversity. How was it measured? As species richness or some other diversity metric? And how were species that were not present in reference sites treated? Some studies exclude "novel" or "invasive" species from such diversity estimates in control-impact studies and this can have major implications for the results.

> You are perfectly right and we are happy to detail and provide more information about local diversity. We included different types of indices used by authors to report alpha diversity (mainly richness, Shannon, Simpson). The type of index was added as a random effect in the models to take their different nature into account. We have clarified this in the Methods section to improve clarity and ensure this is clear to readers.

Manuscript L.324–326

The large majority of studies (99 %) reported local diversity either as richness, Shannon index, or Simpson index (preferentially extracted in this order). A fourth category, "other", regrouped lesser-used diversity indices.

With regard to the second part of your comment, we agree that the choice of the authors to include or exclude some species can have important implications. Our strategy was to include all species if possible and to ensure that the control and treatment were always fully comparable. Therefore, we have consistently prioritised the most "complete" communities in the data extraction process, meaning if results were available for both the "full" community (e.g. including novel species) and a subset of the community (e.g. excluding the novel species), we extracted only the full community. When a study reported only a subset of a community (e.g. without novel species) we included it only if this subset (choice of species inclusion/exclusion) was consistent between the control and the impacted sites (except if the invasive species was the treatment, of course). We clarified this aspect in the Methods section.

Manuscript L.275–279

3) Different definitions of the same community (e.g. including or excluding non-native species). In this case we systematically and consistently prioritised the most "complete" communities (i.e. the definition including the largest diversity). When a study reported a comparison only for a subset of a community (e.g. without non-native species) we included it only if this subset was consistent between the reference and the impacted sites.

1.5. *One smaller issue with the methodology is with the issue of spatial scale. The authors use a few different spatial scale categories, but it was not clear to me what happens when the sites straddle multiple spatial scales. For instance in the Breeding Bird Survey one can have pairs of sites closer than 100km, parts of sites between 100km and 1000km, and pairs greater than 1000km apart.*

> Thank you for this comment, which may be a misunderstanding which we addressed by clarifying the text in the methods section. In our manuscript, the spatial scale is the global geographical extent of the samples, that is the distance between the two furthest reference points within a study, not the variability of scale of distance differences within studies. In short, it is a metric describing if the pairwise distances were on a small or a large scale, per study. We added a sentence to the Methods section to clarify this. It would also be interesting to have information on the within-study variability in distance, yet, unfortunately, we cannot separate the effects by pairwise distances within studies, because the points digitised from the NMDS/PCOA can almost never be individually identified and linked to their corresponding geographical coordinates. Thus, we have to remain with the maximal distance as a proxy of scale.

Manuscript L.289–291

Finally, we classified spatial scales into five categories reflecting the order of magnitude of the geographical extent of the reference sites (i.e. the distance between the two furthest reference sites, as the best proxy of how spatially aggregated/dispersed these communities are).

1.6. *Although Control/Impact (CI) studies are useful, the golden standard are Before/After/Control/Impact (BACI) studies. I think a CI study such as this one stands on its own, but would be interesting to add a couple of sentences on potential limitations of not having the before and after measurements at the impact sites.*

> We agree with you that CI studies are highly useful (and this is reflected in the very large number of published CI studies). We also agree that BACI studies are the gold standard and their respective local finding is for the specific study system the most powerful. However due to their complexity and the need for a reference point in time (Before), the number of BACI studies is much more limited than the CI studies. Of the 2,133 publications included in our meta-analysis, only 93 referred specifically to the BACI design (our selection criteria included CI and BACI studies without discrimination), thus their unfortunately overall relatively low number does not allow for a specific subset meta-analysis. Furthermore, because of their cost and complexity, BACI studies tend to be more prone to geographic and taxonomic biases, thereby limiting their potential for syntheses and meta-analyses (De Palma et al. 2018).

We followed your suggestion to add a few sentences about not having before/after comparisons for impacts.

Manuscript L.196–202

We circumvent these challenges by systematically comparing impacted sites with reference sites (i.e., “Control-Impact” design). While this approach may be in some cases less sensitive than the gold-standard “Before-After Control-Impact” (BACI) design that explicitly accounts for pre-existing differences between the impact and reference groups⁴⁶, it remains by far the most widely used method to measure the real and direct effect of human pressures on biological communities (>95 % of the studies considered had a “Control-Impact” design, and <5% had a BACI design).

De Palma, A. et al. Chapter Four - Challenges With Inferring How Land-Use Affects Terrestrial Biodiversity: Study Design, Time, Space and Synthesis. in *Advances in Ecological Research* (eds. Bohan, D. A., Dumbrell, A. J., Woodward, G. & Jackson, M.) vol. 58 163–199 (Academic Press, 2018).

Specific comments

1.7. I.127-128. Several studies are cited about large-scale biotic homogenisation of exotics, but not the one that arguably did the first global analysis of this kind: Capinha et al. 2015 Science.

> Thank you for the suggestion. We added this important reference.

Manuscript L.130–132

Large-scale biotic homogenisation, when reported, is often linked to the redistribution of species and the facilitation of their dispersal over long distances by humans^{13,25–29}

1.8. I.128-129. I was left wondering if the LRR of studies looking at the pressure invasive species was calculated with all species or just the exotic species.

> The LRR of studies looking at invasive species included all species of the community (both the native and invasive ones). We edited the text to make this clear. Please also see our response to your comment #1.4 and the modification we made to the methods section.

Manuscript L.134–135

In our systematic analysis, we indeed find that invasive species, whose dispersal is typically accelerated by human activities, were the type of pressure with the highest LRR homogeneity of biological communities, that included both native and invasive species (Figure 2a, LRR homogeneity = 0.006, 95% CI = -0.081 to 0.092).

1.9. I.183-185. I find this speculation a bit baseless. How does one go from shifts in community composition to declines in vertebrate populations?

> We think there may have been a misunderstanding of our text. Actually, our intention was to relate declines in vertebrate populations to decrease in local diversity, and not composition. We think that the relationship between population size, risk of local extinction and local diversity is sufficient to support such a discussion/association. We revised the text to ensure that this particular point is clearly articulated.

Manuscript L.191–192

However, contrary to community composition shifts, it is the largest organisms that are experiencing the strongest negative effects of human pressures for local diversity. We speculate that contemporary declines reported in vertebrate populations^{1,43} may be a manifestation of these pressures, given the intrinsic link between population size and risk of local extinction.

1.10. I.215-217 This seems to be overreaching and it's not clear what kind of benchmark the authors want to provide.

> Thank you, we agree with you. We have revised the text of the last sentence to take into account your comment and to clarify the message. Specifically, we toned down the wording and explicitly indicated our intention to provide a benchmark for the development and assessment of future conservation strategies.

Manuscript L.224–227

Our comprehensive analysis provides a highly detailed picture of the state of knowledge available to date on the signal of human impacts on biodiversity and thus provides an important benchmark for the development and assessment of future conservation strategies.

1.11. I.188-189. It's not only about null model expectations. It's about the distribution of the sites themselves and the accuracy of the measurements. See Valdez et al. (2023 Ecography) study showing limitations on the capacity to detect biodiversity trends from time series.

> We agree and added a reference to Valdez et al. 2023, explaining the relevancy of site distribution and accuracy of measurements.

Manuscript L.194–195

These studies are built on time series analysis, which generally lack an impact-reference comparison, may be limited by the number of sites and the accuracy of the measurements⁴⁵, and must be based on adequate null model expectations²³.

Valdez, J. W. et al. The undetectability of global biodiversity trends using local species richness. *Ecography* 2023, e06604 (2023).

Again, many thanks to your constructive and valuable feedback!

Referee #2

I have been specifically asked to review the methodology of meta-analysis. The authors have not followed the recommended procedures for literature reviews and meta-analyses in ecology and conservation (see e.g. Grames and Elphick 2020 Biological Conservation 241: 108385, Foo et al. 2021 Methods in Ecology and Evolution 12: 1705-1720, Nakagawa et al. 2022 Methods in Ecology and Evolution 13: 4-21; Nakagawa et al. 2023 Environmental Evidence 12:8) and as a result multiple details of the review are not clear.

> Thank you for taking the time to review our manuscript and for highlighting specific points in the methodology of the meta-analysis. Following your feedback and suggestions (cf. following comments #2.2–8), we have made significant changes to enhance the clarity of the methodology and we have added a Supplementary Methods section (S3) to provide additional technical and methodological aspects about data collection and extraction. Importantly, however, our meta-analysis goes beyond some standard procedures due to its exceptional nature regarding the number of studies included, scope and effect-sizes (as you noted in comment #2.4; in particular the extraction of all individual data points per study, and not only their mean/variance estimates as is commonly done), which we explain in the following in detail.

Supplementary Information section S3

The metadata of bibliographic references in RIS format and their associated full text in PDF format were centralised and managed in a dedicated Zotero database. The text content of the PDF was extracted using the command-line utility tool pdftotext. The extracted text was analysed to detect any references to PCoA and NMDS analyses by searching for the following regular expression :

```
(?i)NMDS|(?i)PCOA|(?i)principal +coordinate|(?i)multidimensional +scaling
```

The documents containing the target terms were then manually evaluated in order to decide on their inclusion on the basis of a simple rule: presence of at least one biplot graph representing the first two axes of a PCoA or NMDS, and including, while differentiating them, at least two treatments (“control” and “impact”). This procedure was carried out by a single operator. A post-hoc check was carried out on a sub-sample of 100 publications by a second operator. This second operator identified 100% of the publications selected by the main operator (37 in total) while being slightly more conservative by selecting 4 additional references. These references would have been eliminated in any case at the data extraction stage due to the absence of some mandatory information.

The actual data extraction phase was carried out by the 16 project participants. The extraction methodology was communicated to the participants during two workshops (one face-to-face and one online) and through a briefing document. The workshops included a number of practical demonstrations using a variety of examples to review the various extraction stages and special cases. Face-to-face or remote sessions were

then organised, where participants worked together on data extraction and the most complex cases.

The extraction phase was carried out using a dedicated web platform developed specifically for the project. The aim of this platform was to maximise the quality of the user experience while minimising the risk of input errors and offering total control of user privileges to prevent data being exposed. Each participant was provided with a personal account and could add data using an interactive web form. This form supported various features to assist with the data extraction, including required fields, data validation (structures, types, pre-defined options, regular expressions), integration of help in the interface (placeholders and tooltips), and dynamic layout (conditional display of components). The website was built on the content management system Drupal (v.9) and was backed by a MySQL database that connected bibliographic references to data added by users in a seamless and secure way.

a

Add data

Paper selected:

Community-level changes in Australian subalpine vegetation following invasion by the non-native shrub *Cytisus scoparius*

Wearne and Morgan (2004). *Journal Of Vegetation Science*.

b

▼ Record 01

General

Country / Region ? *

e.g. Switzerland; Asia...

Biome ? *

- Select -

Organisms *

- Select -

Figure S7 Screenshots of the interface of the web platform developed for data extraction. Connected users were a, invited to read and extract data from a publication which is randomly selected from the backend database. If they accepted, they were redirected to b, an interactive form designed to assist them with data extraction (only an extract of this form is shown). After they validated the form, data were automatically submitted to the database and users were invited to proceed with a new publication.

2.1. *The authors used Web of Science for literature search and refer to it as a ‘database’ (line 223). WoS is not a database but a platform which provides access to various reference databases. The specific databases and year spans which are accessible through WoS depend on institutional subscription, hence two researchers from different institutions using the same set of keywords in WoS can end up with different number of hits. To make searches repeatable it is important to report which databases within WoS (e.g. Core Collection) and which years where searched.*

> You are absolutely right to note that WoS is a platform providing access to several distinct databases. To ensure clarity and reproducibility, we have specified in the manuscript that we specifically queried the Science Citation Index Expanded (SCI-EXPANDED) provided by Web of Science as part of its Core Collection. Our subscription through ETH-Zurich gives us access to all articles indexed in SCI-EXPANDED and published since 1900, but our search was limited to those published after 1960, as indicated in Extended Data Table 1.

Manuscript L.230–234

To detect studies reporting results of compositional shift and homogenisation in response to human pressures (systematically comparing impact vs. reference scenarios) we conducted an initial bibliographic search on 17/01/2022 on the Science Citation Index Expanded (SCI-EXPANDED) through the Web of Science platform using a combination of 231 relevant keywords (Extended Data Table 1).

2.2. *It is normally recommended to conduct searches for meta-analysis in at least two different databases and compare the results. Why WoS only was used and why was it deemed most appropriate for this study (given the focus on human impact some other databases focusing on more applied research could have been used)?*

> Thank you. We chose to use SCI-EXPANDED because it is a general database and WoS is equipped with the tools and functionalities required for our project (not all the platforms are). For the disciplines we are interested in, the database in question provides access to over 8.5 million documents from more than 4,500 sources. For this reason, it is one of the most widely used databases for meta-analyses in ecology. Managing the 73,000+ results from WoS and obtaining the full texts for them was a complex task. Recommendations to further consult several databases is generally justified when the numbers of entries compared is still overseeable. Yet with most extensive searches and the very broad set of keywords as used here (as detailed in Extended Data Table 1), such an approach becomes unfeasible. While most meta-analyses are doing multiple searches primarily aimed at maximising the number of articles when the questions are specific and the inclusion criteria restricted (which strongly limits the sample size), in our case, we are in the opposite situation as our approach was extremely extensive. Thus, we had to find ways to search and access the data from 73,000+ studied to start with. While highly work-intensive, it ensured we included a very broad set of studies.

Manuscript L.234–235

The SCI-EXPANDED database was chosen for its functionalities, its generalist aspect and its good coverage of targeted disciplines^{49,50}.

2.3. Who did the searchers and study screening checking that the study fits the inclusion criteria? Author contribution statement (lines 425-426) lists the authors involved in data extraction, but it is not clear who did the first two steps on the PRISMA flowchart (publication identification and screening). It is usually recommended that two authors pilot-test the initial screening decision tree and check the degree of agreement between them – has this been done?

> This is a good comment. Actually, the selection of papers was done as follows: First we screened the content of the papers for a list of keywords (totally repeatable, as described in the manuscript and detailed above). Second we reviewed the resulting articles to identify papers that contained possible adequate analyses/data. Third we did the actual extraction of data. For the second step, we intentionally had one main person taking on the role of selecting the papers to avoid inter-person variability. To clarify this, we have now added this important information to the author contribution statement. Yet, as you suggest, we ensured consistency and did a blind parallel extraction “pilot-test” with a second person on a sample of 100 references. This second person selected 100% of the publications selected by the main operator (while including 4 additional). This result confirms how little room there was for subjectivity in the initial selection process and supported our decision to assign this task to a single team member. This information is now added in Supplementary Information S3.

Manuscript L.526–528

FK designed the study, ran the initial search, performed the initial screening, conducted the analyses and led the writing of the paper. TP and FK performed the pilot-test for minimum inclusion criteria.

Supplementary Information section S3

The documents containing the target terms were then manually evaluated in order to decide on their inclusion on the basis of a simple rule: presence of at least one biplot graph representing the first two axes of a PCoA or NMDS, and including, while differentiating them, at least two treatments (“control” and “impact”). This procedure was carried out by a single operator. A post-hoc check was carried out on a sub-sample of 100 publications by a second operator. This second operator identified 100% of the publications selected by the main operator (37 in total) while being slightly more conservative by selecting 4 additional references. These references would have been eliminated in any case at the data extraction stage due to the absence of some

mandatory information (regarding the contrast of control and impact or other criteria used in the extraction step).

2.4. The authors derived their effect sizes from distances measured from PCA and NMDS diagrams. This is not a typical approach to calculating effect sizes in ecological meta-analyses. It is not clear whether the authors have developed this approach by themselves or whether it has been used before in another study. If the former, the justification has to be provided for the new measure of the effect (e.g. it is statistical distribution etc), if the latter, then the references to previous studies using this metric need to be provided. While the extended data Fig 1 is helpful to illustrate how effect size was calculated, I am not clear what is the sample size for this effect – is it the number of pairwise distances? And how the variance for the effect size was computed? Formulas need to be provided.

> Thank you for your comment. We have specifically chosen to use the intermediate information between raw and summarised data that can be extracted from non-constrained analysis graphs because the study of beta-diversity through more classical measures of central tendency is strongly limited by the paucity of studies reporting this information. As we wrote in our response to comment #1.3, where we also elaborate on this, Petsch et al. (2022), for example, could find only 43 studies using beta-diversity metrics to investigate homogenisation driven by invasive species.

The effect sizes we use were described for the first time (including equations) by Zhou et al. (Nature Communications, 2020). As you have correctly understood, the sample sizes are indeed the pairwise distances, since it is these distances that are used to calculate the effect sizes (see eq. 1 and 2 in the revised text), and thus it is based on the extraction of all raw data points in the original studies that allow the direct calculation of all pairwise distances (which is much closer to the original datasets than any extraction of more integrated measures, as commonly done). Note that Zhou et al. derived an equation for the variance of these effect-sizes by directly applying the formula of Hedges et al. (1999) for log-response ratio. In our study, however, the variance of the effect sizes was not computed because we decided to not perform weighted regressions (see our detailed response to your next comment). We have reworded the text to make the reference to the work of Zhou et al. clearer and we have added the equation directly in the Method section for clarification.

Manuscript L.305–316

To do so, we used the community coordinates extracted from the projected plots (PCoA and NMDS) to reconstruct a pairwise Euclidean distance matrix between communities from which we computed two effect sizes, following the approach by Zhou et al.²². First, a measurement of the shift in community composition (LRR shift) estimated as the log-transformed ratio of the mean distance between impacted and reference communities \overline{D}_B and the mean distance within impacted and reference

communities \overline{D}_W , Eq. (1). Second, a measurement of the change in community similarity estimated as the log-transformed ratio of the mean distance within impacted communities \overline{D}_I and the mean distance within reference communities \overline{D}_R . For clarity, this value is multiplied by -1 to express a change in homogeneity (LRR homogeneity) instead of heterogeneity, Eq. (2).

$$(1) \quad LRR \text{ shift} = \ln(D_B/D_W)$$

$$(2) \quad LRR \text{ homogeneity} = -\ln(D_I/D_R)$$

Petsch, D. K., Bertocin, A. P. dos S., Ortega, J. C. G. & Thomaz, S. M. Non-native species drive biotic homogenization, but it depends on the realm, beta diversity facet and study design: a meta-analytic systematic review. *Oikos* 2022, (2022).

Zhou, Z., Wang, C. & Luo, Y. Meta-analysis of the impacts of global change factors on soil microbial diversity and functionality. *Nat Commun* 11, 3072 (2020).

Hedges, L. V., Gurevitch, J. & Curtis, P. S. The meta-analysis of response ratios in experimental ecology. *Ecology* 80, 1150–1156 (1999).

2.5. Meta-analysis has an established methodology (see e.g. Nakagawa et al. 2023 Environmental Evidence 12:8). Effect sizes are normally weighted by the inverse of study variance, it is not clear if any weighting has been done in this analysis and consequently whether different studies had the same or different weight in the analysis. Packages like metafor in R are specifically designed for conducting meta-analyses which allows to implement weighting, fit mixed effects and multilevel models, test for effects of moderators, conduct publication bias tests etc. Why did the authors use instead glmmTNB package which is not specifically suited for meta-analytical models?

> This is a good comment. Actually, we specifically did not weight the studies by their estimated variance to avoid the bias induced by estimating the inverse of the variance. As the sample size is derived from the number of pairwise distances (as indicated to our response to your comment #2.4), it increases with the square of the number of samples/sites (precisely $N * N-1$). This exponential relationship makes weighting by the variance (or by N , see #1.2) completely disproportionate for certain samples (because of the N s in the denominator of the variance formula). With our data, for example, this means that some studies have a weight 150,000 times greater than others, a situation that is clearly neither realistic nor desirable. Recent works have also shown the limits of this approach and the advantages of using an unweighted approach for meta-analyses (Buck et al. 2022 MEE, Hamman et al. 2018 Ecosphere). Given the large size and high heterogeneity of our dataset, we want to ensure equal treatment between studies so as not to draw very broad conclusions that would be the result of a handful of over-represented studies.

Manuscript L.341–350

The models were not weighted to limit the predominance of a few studies on the average effect sizes. As the sample size of LRR homogeneity and LRR shift is the number of pairwise distances between points within each original study and their relationship is exponential, weighting could strongly distort model results. Given the heterogeneity of our data set, there is a risk that uncontrolled variables, irrespective of the number of datapoints, could bias results if the weighting scheme were to exacerbate their weight. Therefore, we adopted an unweighted approach which in this case is a more robust and more conservative strategy. However we show that weighting the models using the square root of the sample size yields qualitatively similar results (Supplementary Information section S5).

Still, based on your comment, we re-estimated the models by weighting the effect sizes by the inverse of their variance in order to compare them with the results presented in the manuscript, see the figures below. We find overall effects in the same direction and of the same order of magnitude with both approaches, as well as comparable patterns in the effect of the different modalities of the moderators tested with a few exceptions which, once again, reflect the fact that a handful of studies are over-represented. Note the much smaller sample size for alpha diversity because a lot of standard errors could not be found/extracted (another reason for us to favour unweighted regressions).

Figure S10 Impacts of human pressures on homogeneity and shifts in composition of biological communities estimated using weighted regressions. The weighting scheme applied is the inverse of the square root of the variance of the effect sizes, computed following Zhou et al.1 . a, Log-response ratio of community homogeneity (logarithm-transformed ratio of impacted to reference values, LRR homogeneity). b, Log-response ratio of community composition shift (LRR shift). The global response (all data) is shown on the first row of each panel and is separated by factors in the following rows. The numbers between parentheses indicate the number of comparisons. For each category the dot represents the marginal mean computed from the model; dot size is proportional to number of studies included. The larger bar shows the 95 % confidence interval and the thinner bar represents the 99 % confidence interval.

Figure S11 Impacts of human pressures on local diversity estimated from experimental studies using weighted regression. The weighting scheme applied is inverse of the variance of the effect size, computed following Hedges et al.2 . Log-response ratio of local diversity (logarithm-transformed ratio of impacted to reference values, LRR local diversity). The global response (all data) shown on the first row is separated by factors in the following rows. The numbers between parentheses indicate the number of comparisons. For each category the dot represents the marginal mean computed from the model; dot size is proportional to number of studies included. The larger bar shows the 95 % confidence interval and the thinner bar represents the 99 % confidence interval.

Regarding the choice of software, we readily acknowledge the value of specialised packages for conducting meta-analyses. However, as our study is not a standard meta-analysis, we wanted to opt for a more modular and flexible approach. The vast majority of models offered by a package such as metafor are generalised linear mixed-effects models. The glmmTMB package provides a state-of-the-art implementation of GLMMs, probably one of the most reliable and most complete available in the R ecosystem. It is interesting to note that metafor actually does not do the heavy lifting when it comes to fitting GLMMs, but relies on the functions provided by glmer and glmmTMB (at the user's choice, according to the documentation).

Hamman, E. A., Pappalardo, P., Bence, J. R., Peacor, S. D. & Osenberg, C. W. Bias in meta-analyses using Hedges' d. *Ecosphere* 9, e02419 (2018).

Buck, R. J., Fieberg, J. & Larkin, D. J. The use of weighted averages of Hedges' d in meta-analysis: Is it worth it? *Methods in Ecology and Evolution* 13, 1093–1105 (2022).

2.6. Multiple records were extracted per study (243-244) which might make effect sizes non-independent of each other. Publications were included as a random factor in the models (line 293) presumably to control for this non-independence. A better approach would be to use a multi-level model with study ID and effect ID as random factors which would allow to include both between- and within- study effects as in Nakagawa et al. 2023 *Environmental Evidence* 12:8)

> Thank you for this suggestion. We fully recognize the value and usefulness of three-level meta-analysis models. However, in our situation, the within-study replication rate is remarkably low (on average 1.72 effects/study, we now added this information to the manuscript). This means that within-study replication is likely to play a very negligible role here (as there is very little within-study effect-replication). The specific highly independent structure of our data actually also makes hierarchical models difficult, if not impossible to fit, due to overparameterization (consequently, we do not suffer from some within-study replication issues inherent to most meta-analysis, but could also not use the respective suggested analytical approaches to such datasets). As a result, our tests with different solvers systematically lead to convergence errors on three-level models. Consequently, simpler mixed effects models dealing with between-study heterogeneity but without the effect ID as a nested random factor are sufficient and most suitable here for three reasons:

1) As indicated, the within-study replication rate is very low (less than 2 effects per study, on average).

2) Even when multiple effect-sizes are extracted within a study, our selection strategy described L243–255 prevents within-study pseudo-replication; i.e. when a study reports several effect-sizes, this is most systematically for a different combination of our fixed effects (already explicitly taken into account in the model).

3) As a consequence of 1) and 2), for the hierarchical models we were able to fit without convergence error, the difference with the non-hierarchical mixed model appeared to be negligible (parameter values varied after the third decimal place).

We clarified this methodological choice in the manuscript.

Manuscript L.339–341

In all models, publication was used as a random effect to account for between-study heterogeneity. Within-study heterogeneity was not explicitly included because the number of extracted records per study was only 1.72, on average.

2.7. Why were moderators fitted one at the time (lines 288-289) instead of together (and thus allowing to test for interactions between them)? Has the potential confounding between the moderators been tested?

> To test the effect of moderators, they were fitted together in the different models precisely to incorporate their potential to confound each other. Consequently, all the tests of moderators (Type II Wald χ^2 tests) presented in the main text are based on complete models incorporating all moderators (as fixed effects). In parallel, we generated different models by fitting the moderators separately, in order to generate Figure 2 and Figure 3. The aim is to simplify the readability and interpretation of these plots, since in this case the parameters are "averages" unconditioned by the other moderators (but corrected for pseudoreplication). It seems that our explanations were not sufficiently clear, so we have reworked the text to ensure clarity and remove any possible ambiguity.

Manuscript L.333–339

For LRR homogeneity and LRR shift, models included **all the moderators together** as fixed effects and two random effects (intercepts): publications and type of study (field observation, field manipulation, or experimental). For LRR local diversity the model included **all the moderators together as fixed effects** and three random intercepts: publications, type of **local diversity index (richness, Shannon index, Simpson index, or other)**, and type of study (field observation, field manipulation, or experimental).

Manuscript L.352–357

Additionally, for each response variable (LRR homogeneity, LRR shift, and LRR local diversity), we fitted four independent mixed models including **only one moderator at once** and the same random effects as described above. From each model we computed estimated marginal means and their 95 % and 99 % confidence interval, respectively, for each level of the included factor. Estimated marginal means are an efficient approach to **decompose** the effect of a specific factor **and they were used to produce Figure 2 and 3.**

2.8. *Publication bias testing is an important part of the meta-analysis (Nakagawa et al. 2022 Methods in Ecology and Evolution 13: 4-21), particularly if it includes only published literature as the given study. No tests for publication bias appear to be conducted.*

> Thank you very much, we happily added this analysis on publication bias. We added a paragraph in the Methods section as well as a supplement (Supplementary information S6) with the detailed results of our tests.

We combined a visual inspection of the funnel plots with a file drawer analysis and a P-curve analysis. While there is an asymmetry in the funnel plot of LRR shift, it is not directly caused by small studies. Moreover, the Fail-Safe N and the test for p-values right-skewness (evidential values) demonstrate that the observed effects are robust to publication bias.

Manuscript L.366–370

We assessed publication bias by inspecting funnel plots for potential asymmetry driven by small-studies⁵⁴. Additionally, we performed a File Drawer Analysis^{55,56} and a P-Curve analysis⁵⁷ to further confirm the robustness of our findings against selective publication. The results of these analyses (Supplementary Information section S6) suggest that our findings are robust and not unduly influenced by publication bias.

Supplementary Information section S6

We assessed potential publication bias for each outcome measure, namely LRR homogeneity, LRR shift and LRR local diversity. Our strategy was based on a combination of funnel plot visual inspection³, file drawer analysis⁴ and P-curve analysis⁵. We observe no evidence of asymmetry on the funnel plots with regard to LRR homogeneity (Figure S1a) and LRR local diversity (Figure S1c) while for LRR shift a small asymmetry can be observed (Figure S1b). This asymmetry, however, appears to be linked to the effect size distribution and has no apparent relationship with the standard error. A small-study effect, on the contrary, would have resulted in an over-representation of studies in the lower right-hand part of the funnel plot only.

To ascertain the robustness of our findings against possible publication bias, we performed a File Drawer Analysis and calculated the Fail-Safe N, a measure estimating the number of additional studies with null findings required to render observed effects non-significant. The File Drawer Analysis was conducted with the function `fsn` from the R package `metafor` using the generalised method of Orwin and Rosenberg^{4,6}. We found Fail-Safe N values of 2,757,894 for LRR shift and 14,410 for LRR local diversity. LRR homogeneity was not tested as its grand mean was not found significant in our analyses. Overall these large to very large values (5 to 1000 times more studies than included in our analyses) indicate that our results are highly robust to publication bias.

Finally, we performed a test for p-values right-skewness (P-Curve analysis) to further validate the robustness of our results, specifically with regards to p-hacking. For every tested effect sizes, we found that both the half and full p-curve tests clearly indicated a right-skewed shape: LRR homogeneity (Full: $Z = -22.7$, $P < 0.001$, Half: $Z = -21$, $P < 0.001$), LRR shift (Full: $Z = -61.8$, $P < 0.001$, Half: $Z = -51$, $P < 0.001$) and LRR local diversity (Full: $Z = -69.5$, $P < 0.001$, Half: $Z = -70.1$, $P < 0.001$). Together these results indicate that there is strong evidential value behind our data.

Figure S12 Funnel plots of effect sizes versus standard error for a. LRR homogeneity, b. LRR shift and c. LRR local diversity.

Referee #3

This manuscript describes an ambitious effort to synthesise evidence of how the major direct drivers of biodiversity change affect three important aspects of the composition of ecological communities. They estimate each driver's (a) net effect on local alpha diversity; (b) effect on compositional change from a reference community, used in lieu of pre-impact communities; and (c) effect on beta diversity, as a way of testing whether the driver causes biotic homogenization or differentiation. They also test whether effects differ among five drivers, nine major taxonomic groups, five different spatial scales, and three 'main biomes' (terrestrial, freshwater and marine). The manuscript is generally very clear, well written and well presented. The logic behind the analysis is laid out very clearly, and Figure 1 is a thing of beauty.

The main results can be summarised as follows. (a) Drivers reduces local diversity significantly, by an average of 13.3% (back-transforming the effect size on line 178), with effect sizes differing among taxonomic groups and drivers. Indeed, pooled effect sizes for resource exploitation and climate change are smaller than those for other drivers and non-significant (Figure 3). (b) All drivers are inferred to cause significant shifts in community composition, in all groups and in all major biomes, with the effects being more pronounced as spatial scale decreases (Figure 3). (c) Effects on community homogeneity are smaller and more complex to summarise: differentiation at smaller spatial scales tends towards homogenization at larger scales, and different taxonomic groups show different tendencies.

Thank you for your supportive review! We particularly appreciate your positive comments on the high quality of the figures and presentation of the results, as we indeed have made a considerable effort to do this well. We are thus happy to see that we have reached this goal.

We appreciate your highly constructive comments. We addressed and incorporated all of them accordingly, see below our responses in detail.

3.1. There is a lot of novelty here. Although previous syntheses, including some in Nature, have shown effects of one or more drivers on local diversity or community composition, I am not aware of any paper that matches the ambition and breadth of this one – with meaningfully large sample sizes for terrestrial, freshwater and marine communities, for each driver, and for such a diversity of taxa. There is an elegance to the analysis, with scale, taxon, driver and 'main biome' all considered simultaneously.

The significance of the paper is less clear. The authors are right not to emphasise the differences in effect sizes among drivers, because dose-response relationships depend on the dose as well as the effect-per-unit-dose, but it does limit the broader impact of the results. I suspect this limitation is why the manuscript is largely framed around the novelty of considering both compositional change and homogenization/differentiation together (lines 73-79), which I did not find a particularly strong motivation.

> We really appreciate your recognition of the novelty, ambition and breadth of our analysis and findings and that there is no other comparable synthesis.

We agree with you that a central aspect of our work is the integration of drivers, taxonomic groups and multiple spatial scales across terrestrial, aquatic, and marine biomes. We think that this plurality of aspects is actually the core and main significance of our manuscript. We are not specifically seeking to establish a hierarchy between anthropogenic pressures. Our main message is instead that while the different types of drivers cannot be directly compared, they are all important. For example, Bellard et al. (2022, Nat. Comm.) specifically warned against the temptation to rank biodiversity threats according to a hierarchy that is strongly dependent on the context and the metrics employed, and we do—also in line of parsimony and interpretations only based on quantitative facts—agree with this.

The other risk is to produce an oversimplified image for the general public, the media and policy-makers that would likely prove to be counterproductive in the fight against biodiversity erosion. On the contrary, we believe that our results, which include the main drivers of this erosion in one of the most ambitious analyses ever carried out of human impacts on biodiversity, are a unique opportunity to illustrate the ideas of Bellard et al. (this reference was added to the manuscript) and to reiterate the importance of taking all fronts into account to tackle the biodiversity crisis.

Manuscript L.69–81

Crucially, previous research attempting to generalise biodiversity change have hitherto neglected two key elements. Firstly, most past studies looked at biodiversity change across time using individual time-series and did not contrast findings to reference controls⁹. Secondly, previous studies have rarely differentiated between changes in local diversity versus changes in variation in diversity across space. Unfortunately, the studies that have integrated these elements are generally restricted to a certain type of pressure or to a particular biome^{15–19}. Consequently, we lack generalisations on the effects of human pressures on ecosystems and our understanding of biodiversity change with regard to its different dimensions remains incomplete. Thus, we are limited in our capacity to disentangle its underlying drivers. Given the multifaceted aspect of biodiversity, and the plurality of drivers, organisms and spatial scales, the current lack of synthetic understanding and attribution of general impacts of human pressures on biological communities is hindering adequate actions and mitigation strategies^{20,21}.

Manuscript L.88–95

We manually and systematically extracted data points from these ordination plots, each of these points representing the composition of an individual biological community (Extended Data Fig. 1). This meta-analytical framework, first introduced by Zhou et al.²², allowed us to discriminate between changes of homogeneity and shifts in composition of biological communities in relation to human pressures. In contrast to previous studies mostly restricted to biomonitoring time series, we focused on direct impact studies, considering any of the five most predominant anthropogenic pressures:

land-use change, resource exploitation, pollution, climate change and invasive species^{1,6}.

Manuscript L.156–159

Our results show **that** these human pressures **systematically change the composition of communities** and provide critical insights on the magnitude of effects across human pressures, supporting the notion that all human pressures need to be considered when attempting to bend the curve on biodiversity loss³⁷.

Manuscript L.221–227

By systematically assessing how the five major global human pressures impact biodiversity, we quantitatively attribute biodiversity change in impacted versus reference communities, integrating effects on both local diversity change and changes in community composition. Our comprehensive analysis **provides a highly detailed picture** of the state of knowledge available to date on the signal of human impacts on biodiversity **and is thus an important benchmark for the development and assessment of future conservation strategies.**

Bellard, C., Marino, C. & Courchamp, F. Ranking threats to biodiversity and why it doesn't matter. Nat Commun 13, 2616 (2022).

I also have two fundamental concerns with the data and analysis in the manuscript. One is easy to fix but might change the conclusions relating to all three main results; the other might not be fixable but may simply invalidate results (b) and (c) and all conclusions drawn from them.

3.2. First, there is wide recognition that meta-analysis of published literature must consider the possibility of publication bias, i.e., the under-reporting of non-significant effect sizes. The complete omission of any mention of publication bias from this manuscript is therefore unacceptable. It should be tested for using an appropriate method and the results, implications and limitations of the test reported.

> You are absolutely right. As we have also indicated in our reply to reviewer 2 (#2.8), we have now added a full analysis of possible publication bias (see Supplementary Information section S6). Through this analysis, we show that publication bias is firstly very limited and, secondly, that its potential impact on our results is very low and thus not a cause for concern. Specifically, we combined a visual inspection of the funnel plots with a file drawer analysis and a P-curve analysis. While there is an asymmetry in the funnel plot of LRR shift, it is not directly caused by small studies. Moreover, the Fail-Safe N and the test for p-values right-skewness (evidential values) demonstrate that the observed effects are robust to publication bias. We have added a paragraph to the main text where we describe this analysis of publication bias we carried out, outline our findings, and refer readers to Supplement S6 for further information.

Manuscript L.366–370

We assessed publication bias by inspecting funnel plots for potential asymmetry driven by small-studies⁵⁴. Additionally, we performed a File Drawer Analysis^{55,56} and a P-Curve analysis⁵⁷ to further confirm the robustness of our findings against selective publication. The results of these analyses (Supplementary Information section S6) suggest that our findings are robust and not unduly influenced by publication bias.

Supplementary Information section S6

We assessed potential publication bias for each outcome measure, namely LRR homogeneity, LRR shift and LRR local diversity. Our strategy was based on a combination of funnel plot visual inspection³, file drawer analysis⁴ and P-curve analysis⁵. We observe no evidence of asymmetry on the funnel plots with regard to LRR homogeneity (Figure S1a) and LRR local diversity (Figure S1c) while for LRR shift a small asymmetry can be observed (Figure S1b). This asymmetry, however, appears to be linked to the effect size distribution and has no apparent relationship with the standard error. A small-study effect, on the contrary, would have resulted in an over-representation of studies in the lower right-hand part of the funnel plot only.

To ascertain the robustness of our findings against possible publication bias, we performed a File Drawer Analysis and calculated the Fail-Safe N, a measure estimating the number of additional studies with null findings required to render observed effects non-significant. The File Drawer Analysis was conducted with the function `fsn` from the R package `metafor` using the generalised method of Orwin and Rosenberg^{4,6}. We found Fail-Safe N values of 2,757,894 for LRR shift and 14,410 for LRR local diversity. LRR homogeneity was not tested as its grand mean was not found significant in our analyses. Overall these large to very large values (5 to 1000 times more studies than included in our analyses) indicate that our results are highly robust to publication bias.

Finally, we performed a test for p-values right-skewness (P-Curve analysis) to further validate the robustness of our results, specifically with regards to p-hacking. For every tested effect sizes, we found that both the half and full p-curve tests clearly indicated a right-skewed shape: LRR homogeneity (Full: $Z = -22.7$, $P < 0.001$, Half: $Z = -21$, $P < 0.001$), LRR shift (Full: $Z = -61.8$, $P < 0.001$, Half: $Z = -51$, $P < 0.001$) and LRR local diversity (Full: $Z = -69.5$, $P < 0.001$, Half: $Z = -70.1$, $P < 0.001$). Together these results indicate that there is strong evidential value behind our data.

Figure S12 Funnel plots of effect sizes versus standard error for a. LRR homogeneity, b. LRR shift and c. LRR local diversity.

We also would like to highlight that it is an inherent aspect of any meta-analysis that it can only cover what has been published. Given the very broad number and types of papers included in our analysis and the long time-period covered, we are confident that publication bias is much less of an issue than in any other meta-analysis that covers shorter time periods and more specific topics.

3.3. *Second, and more problematically, I think that a large proportion of the results in (b) and (c) could well be statistical artifacts caused by a ubiquitous ecological phenomenon that the authors mention (lines 134-136) but whose implications for their analysis they do not think through: the 'distance-decay' relationship whereby compositional similarity between two communities falls (monotonically and fairly predictably) with the distance between them, other things being equal.*

The authors have gone to admirable and exemplary lengths to capture the compositional distance between the communities at their sites, digitising each site's location in a suitable ordination plot and thence inferring the compositional distance matrix linking all sites. The problem is that geographic distance between sites – which predicts compositional distance so well that it is often used as the starting point for the scaling optimisation – is not considered or controlled for but (crucially) is very likely to differ systematically in ways that bias results (c) and (especially) (b).

Result (b) depends on the untested (and unstated!) assumption that the average geographic distance between an impacted site and a reference site is the same as the average geographic distance between two sites in the same category. This is very unlikely to be met. Studies commonly have multiple sampling points close together within a patch that is treated homogeneously. Sometimes this is pseudoreplication that goes entirely unrecognised through analysis and peer review (the effective sample size might really be one comparison between a block of impacted sites and a block of reference sites), or sometimes there is a higher-level structure to the design (with the pseudoreplicated samples being pooled prior to testing hypotheses). Whenever this happens, the average within-condition distance will be less – often much less – than the average between condition distance. (Conversely, if the design has spatial blocks each with only one reference and one impacted site, the bias will be the other way; my experience is that such designs are rarer however.)

Result (c) depends just as strongly on the untested assumption that the average geographic distance between two impacted sites is the same as the average geographic distance between two reference sites. This assumption will also be violated under many commonly used study designs. A common design in heavily-impacted settings is that the same patch of reference habitat is used to make multiple reference-impact comparisons, using paired comparisons of sites within each of which a different site just inside the reference patch is compared with a nearby site just outside. In such a design, the mean reference-reference geographic distance is smaller than the mean impact-impact difference, biasing the inference towards differentiation. Likely circumstances for this kind of design would be studies assessing effects of protected areas (where resource exploitation should be lower) on community composition.

I see two possible ways forward with this problem. First, it may be possible, for at least a subset of studies, to use maps or tables of geographic coordinates to compare the three kinds of geographic distances. My objection obviously falls if there is no bias or if the bias opposes rather than drives the results, and I would be supportive of publication in Nature. Second, the issue of distance-decay needs to be foregrounded and the resulting limitations given prominence. There would still be enough novelty, ambition and quality for the resulting

manuscript to be published in an excellent journal in that case, but I don't think it would be competitive for Nature.

> Thank you for this detailed comment and for taking the time to elaborate further on this important point. While we agree that the design choices made by the authors of the studies included in our meta-analysis do not necessarily correspond to a purely random distribution in space, we respectfully disagree that the observed patterns are due to statistical artefacts. As you rightly pointed out, ecological studies have to deal with the complexity of the field. Nevertheless, we defend the idea that the presence of spatial bias in site location, if non-systematic and moderate (we will come back to this below), does not call into question our approach or our conclusions.

Firstly, we would like to emphasise the foundational principle of all meta-analyses, namely that they all rely on the assumption that the published literature is reliable and reflects the scientific findings within a given field. While more sophisticated approaches exist (e.g. before-after control-impact (BACI), cf. response to comment #1.6), control-impact study designs (CI) are by far the most widely used method for studying the impact of human activity on biodiversity. The CI approach has been extensively tested and validated. Although it is not perfect, site distribution is an issue only if it ignores the contextual elements (species, ecosystem, time and spatial scales, etc.) of the study, which is unlikely. More sophisticated study designs, such as the BACI design are massively less commonly used (<5% of all our studies have such a design), which unfortunately does not allow us to solely focus on those without a massive loss of information.

In accordance with the aforementioned foundational principle of meta-analysis, we are confident and assume that the authors of all the respective included papers are experts in their respective fields and possess an intimate understanding of the environments and ecosystems they investigate. As such, we think these original study authors are in the best position to make informed decisions regarding study design and site selection. Further, we would like to stress that ensuring comparability between treatment groups extends beyond merely controlling for geographical distances; it necessitates careful consideration of various contextual elements, including but not limited to the types of organisms studied, their dispersal abilities, and the spatial scale within and between the treatments. For example a sampling design that would separate control and impact sites in two completely distinct clusters will have very different implications if the clusters are separated by 50 m or 50 km, or if the focal species are microbes or large vertebrates. This is because the distance-decay relationship can show a wide range of shapes, depending on many contextual factors (Clark et al. 2021; Graco-Roza et al. 2022). Moreover, distance-decay relationships are typically reported and characterised in the context of biogeography, where the scales are much larger than the potential geographical bias we are talking about here. We trust that the authors have diligently accounted for these factors in their methodological choices. We also wish to highlight that all the studies included in our analysis went through peer-review. While focusing solely on the literature published in scientific journals and excluding grey literature can have its drawbacks (see comment #2.8), in this case it guarantees that the methodological choices have passed the peer validation stage. Spatial sampling design, including considerations of comparability and independence, is a central aspect of control-impact studies and this point is closely scrutinised by the reviewers, ensuring that studies meet minimal methodological standards.

With that said, we have carefully considered your suggestions for remedying the problem and we particularly appreciate that you have taken the time to point us in these possible directions. We felt that your request to extract the geographical coordinates of the sampling points for a subset of our studies was fair and would be a welcome addition to help the reader understand the characteristics of our dataset. We therefore extracted the geographical coordinates from the maps published in a subset of 200 articles, using the same digitising technique as described in the manuscript for extracting beta-diversity data from projection plots. We also used the same effect-sizes (LRR homogeneity and LRR shift) this time applied to the geographical distance matrices to analyse whether 1) the distances between samples subjected to the same treatment were equivalent and 2) whether the distances between different treatments were different than the distances within treatments.

Our methodology and our findings are presented in the Supplementary information S7. Overall, we found significant spatial LRR homogeneity and shift, which matches your expectations. However, we reiterate that this does not compromise the comparability between Control and Impact treatments, and we would like to draw your attention to the distribution and mean of these values, which remain very small overall. We found a positive average LRR homogeneity value (~ 0.17), which seems to indicate that the impacted sites are minimally closer together than the reference sites. However, the distribution shows a variability of situations (going in both directions) and the mean indicates that this effect remains very moderate overall. The LRR shift has a higher mean value (~ 0.32) and its distribution is a bit asymmetric with a handful of studies showing clear signs of spatial clustering, but again, this is not saying if the absolute spatial distances, the landscape connectivity and species dispersal capacities are compatible with a perceptible effect of the 'distance-decay' relationship that would undermine comparability. Interestingly we often observed two-level designs with pairwise CI comparisons at local scale replicated at larger scale to limit this shift effect.

Most importantly and still for this subset of studies, LRR homogeneity based on community is poorly and not significantly correlated to LRR homogeneity based on spatial distances, and the same applies for LRR shift, hence supporting that the community patterns reported in the manuscript are independent from the spatial distribution of sites. We believe that this is the result and concrete manifestation of all the aforementioned points.

Finally, we would like to point out that our assessment of spatial distances primarily involved field studies (and the respective geographical maps provided in such studies). Experimental studies, which often feature highly controlled designs, form a significant portion of our main analysis. We are confident that these studies employ robust blocking designs—regarding spatial distributions—that effectively minimise spatial biases. The beta-diversity patterns observed in these experimental studies align closely with those seen in the global dataset (see Figure below). However, experimental studies could not be included in our geographical analysis since they virtually never provide a map or schematic diagram showing the spatial distribution of the samples.

In conclusion, while we acknowledge that some spatial dependencies in site location cannot be completely excluded, we also reiterate that 1) this is something the individual studies must have already addressed (and it is highly unlikely they all introduced the same systematic bias, as also corroborated by our additional analyses on a subset of studies) and 2) that some spatial dependencies between impact and control may be actually inherent to

anthropogenic impact, thus being a true and actual signature of this very same impact. As such, we maintain confidence in the validity of our findings, which are based on a comprehensive synthesis of peer-reviewed studies that we believe were conducted with meticulous attention to methodology.

We truly appreciate your thoughtful comments and have taken all steps to clarify the rationale and considerations underlying our analysis, both in the manuscript and the Supplementary Information.

Manuscript L.371–376

Additionally, we analysed the spatial layout of reference and impacted sites across a subset of studies ($n = 200$), specifically investigating differences between within-treatment distances (ie. spatial homogeneity) and between within and between treatment distances (ie. spatial shift). We found limited but significant evidence of spatial patterns in site distribution. However these patterns did not correlate with the changes observed in community composition between reference and impacted sites (Supplementary Information section S7).

Supplementary Information section S7

We analysed the spatial distribution of reference and impacted study sites in order to compare geographical patterns with those observed in biological communities.

To this end, we extracted the geographical coordinates of reference and impacted sites for a subset of 200 articles. The coordinates of impacted sites were extracted directly from the maps published in the articles, using the same digitising technique described in the manuscript for biological communities. We also used the same effect sizes (LRR homogeneity and LRR shift) here applied to geographic distance matrices to analyse whether 1) distances between samples within treatments were equivalent (LRR homogeneity) and 2) whether distances between treatments were greater than distances within treatments (LRR shift). Spatial LRR homogeneity exhibited a distribution closely centred around zero (Figure S10a) and a small (0.17) yet significant mean value ($t = 2.979$, $df = 199$, $P = 0.003$). Similarly we found the mean of spatial LRR shift (0.32) significant ($t = 7.087$, $df = 199$, $P < 0.001$) and its distribution right skewed (Figure S10b). However, neither of the two space-based effect-sizes seems clearly correlated with their equivalent community-based measures (LRR homogeneity: $\rho = 0.08$, $t = 1.142$, $df = 198$, $P = 0.25$, Figure S10c; LRR shift: $\rho = 0.13$, $t = 1.888$, $df = 198$, $P = 0.06$, Figure S10d).

Figure S13 Distributions of a. LRR homogeneity and b. LRR shift values computed from spatial distances for a subset of 186 studies. Relationship between c. LRR homogeneity and d. LRR shift values computed from spatial and community distances.

Graco-Roza, C. et al. Distance decay 2.0 – A global synthesis of taxonomic and functional turnover in ecological communities. *Global Ecology and Biogeography* 31, 1399–1421 (2022).

Clark, D. R., Underwood, G. J. C., McGenity, T. J. & Dumbrell, A. J. What drives study-dependent differences in distance–decay relationships of microbial communities? *Global Ecology and Biogeography* 30, 811–825 (2021).

I also have a few very much less fundamental suggestions that the authors may find helpful for improving the manuscript. In manuscript order, these are:

3.4. Lines 67-76 imply (perhaps unwittingly) that the data are all time series from matched sets of reference and impacted sites but, as lines 254-255 make clear, time series were not used (in that only the last year of them was used). Please clarify.

> It seems that we could have been clearer on line 67–76, since this paragraph does not relate to our analysis but to past analyses, many of which have been based mainly on time

series data. We have edited this section to clarify. Thank you for pointing this out.

Manuscript L.70–77

Firstly, most past studies looked at biodiversity change across time using individual time-series and did not contrast findings to reference controls⁹. Secondly, previous studies have rarely differentiated between changes in local diversity versus changes in variation in diversity across space. Unfortunately, the studies that have integrated these elements are generally restricted to a certain type of pressure or to a particular biome^{15–19}. Consequently, we lack generalisations on the effects of human pressures on ecosystems and our understanding of biodiversity change with regard to its different dimensions remains incomplete.

3.5. Lines 234-237: I briefly thought that this sentence included an absolutely fatal flaw but now think that's a misreading of an ambiguous statement. My (hopefully) mistaken reading was that you discarded sources where the reference and impacted groups of sites were not different ("could not be visually separated"), but I assume that you meant simply that the plot had to differentiate sites from the two conditions by colour or plotting symbol. But please clarify so no-one else leaps to the same wrong conclusion.

> Thank you for noticing this ambiguity. Indeed, this is the second option. We have modified the text to ensure this is clear.

Manuscript L.251–252

The plot had to report at least two groups, the reference (least impacted) and the impacted groups, which could be visually separated by distinct colours or plotting symbols, respectively.

3.6. Lines 269-271: A range of different diversity measures were used. Although a random intercept was fitted to accommodate heterogeneity arising from this range, did you use any one particular 'version' of a diversity index in preference to others? (e.g. consider the range of ways Simpson's index gets presented.)

> Yes we did. Diversity measures were extracted in 4 different categories: richness, Shannon index, Simpson index, or other (to group together less frequently used indices). When several indices were provided we choose one, following this order of preference. We have added this information to the Methods section to clarify.

Manuscript L.323–326

For each record, the mean local diversity for the impacted and reference communities was also recorded (if reported). The large majority of studies (99 %) reported local

diversity either as richness, Shannon index, or Simpson index (preferentially extracted in this order). A fourth category, "other", regrouped lesser-used diversity indices.

Referee #4

This paper presents a meta-analysis of global studies on the biodiversity impacts of five important aspects of human action: habitat alteration, resource exploitation, pollution, climate change and invasive alien species. It brings together data from a wide range of experimental and observational studies, but restricts itself to those that explicitly compare impacted and (relatively) pristine sites, and specifically to those that consider community change using one of two ordination methods. The result is the sort of broad-spectrum, general interest human impact assessment that is likely to garner wide interest.

> Thank you very much for your review! We truly appreciate your recognition of the value and potential large impact of our study, as well as your thoughtful and constructive comments. Below, we respond point by point to each of your specific comments (we have addressed and incorporated all of them accordingly, improving the clarity of our manuscript).

4.1. I am not a specialist on the particular methodology employed, but it seemed largely sensible. I would advise enlisting a review from someone steeped in this approach (I will make recommendations in comments to the Editor). I was, however, curious about the choice of ordination methods to include here; while PCoA and NMDS are widely used, many other methods, such as DCA (or DECORANA), are also widely used in the literature; little justification is given for why the two focal measures are the only ones appropriate here. Also, despite frequent mentions of assessing "Local diversity", I was unable to determine whether the authors meant raw species richness, some diversity index (Shannon? Simpson?) or something else (Chao index? Rarefied richness?). The Methods section (lines 269-270) indicates that some sort of "diversity index" is used, but perhaps that means that different studies were assessed using different measures (if so, how were these differences reconciled?) It was also not clarified whether biodiversity impacts of invasive species counted the alien species themselves in the diversity measures.

> We agree with you that there are several other methods to ordinate community composition in two dimensions, such as PCA, CA, DCA, etc. Importantly, we specifically focused on PCoA and NMDS here because these methods are distance-based methods and therefore directly relate to the concept of beta diversity (ie., the differentiation among local sites), while PCA, CA, DCA, etc. rather focus on summarising and maximising the variance or correspondence structure of the data, which may not completely align with the ecological distances present in the data. In contrast, NMDS and PCoA explicitly aim to represent and preserve the pairwise dissimilarities between samples in the new reduced-dimensional space, which is our primary interest here. We edited the text to clarify this methodological choice.

Manuscript L.241–245

NMDS and PCoA explicitly aim to represent and preserve the pairwise dissimilarities between samples in the new reduced-dimensional space. Therefore, they were preferred to other ordination methods such as Principal Component Analysis or

(Detrended) Correspondence Analysis whose focus is first on summarising and maximising the variance or correspondence structure of the data⁵¹.

Regarding local diversity, you are right to point out that some additional information would be helpful (also noted by reviewer 1 in #1.4). In our analyses, we included different types of alpha diversity indices commonly used in publications, mainly richness, Shannon and Simpson indices, plus an “other” category to regroup less commonly used indices. We added this information in the text of the Methods section, and also indicated that the type of index was added as a random effect in the meta-analytical models to take their different nature into account. Also, as we replied to reviewer 1 (response #1.4), we prioritised the most “complete” communities (i.e. including novel species) in the data extraction process and in the case of a subset of a community (e.g. without novel species) we included it only if it was consistent between the control and the impacted sites (except if the invasive species was the treatment). We appreciate you pointing out that this information was lacking. We agree and we have added it to the Methods section.

Manuscript L.323–326

For each record, the mean local diversity for the impacted and reference communities was also recorded (if reported). The large majority of studies (99 %) reported local diversity either as richness, Shannon index, or Simpson index (preferentially extracted in this order). A fourth category, “other”, regrouped lesser-used diversity indices.

Manuscript L.277–279

When a study reported a comparison only for a subset of a community (e.g. without non-native species) we included it only if this subset was consistent between the reference and the impacted sites.

4.2. The paper's findings were interesting, and mostly in line with the scientific consensus. It is certainly not surprising to find that the various human interventions significantly shift community composition, and that most have adverse effects on local diversity (whatever that means here). The authors suggest (lines 153-154) that the community changes seen are in "a consistent direction" -- but no detail is given as to the nature of that directional change, nor whether it is consistent across drivers. The most striking finding, however, was a lack of impact on biotic homogeneity (here operationally defined as scatter on an ordination plot). It was also interesting (but little mentioned in the text) that climate change impacts on diversity appeared quite weak, but of course it is difficult to find appropriate "control" sites for comparisons in this case.

> Thank you for your comment, we are really pleased that you found our study and results interesting! The consistent direction referred to (L153–154) relates to community compositional changes. Given that the biplot graphs extracted are not oriented and that we

do not know the underlying composition of the biological communities from which we extracted the coordinates, it is difficult to talk about a direction of change in the strict sense. It seems that our formulation was ambiguous and unclear, since our intention here was rather to indicate that changes in community composition are systematic between drivers. We have edited the text accordingly.

Manuscript L.156–158

Our results show that these human pressures systematically change the composition of communities and provide critical insights on the magnitude of effects across human pressures [...]

Regarding climate change, as you are right to point out, it is difficult to find appropriate control sites for comparisons (besides experimental conditions). More generally, and in line with Bellard et al. (2022, Nat. Comm.), we prefer to remain cautious about the hierarchy between human drivers because direct comparison is difficult as we explain in the text and in our response to reviewer 3 (comment #3.1).

Bellard, C., Marino, C. & Courchamp, F. Ranking threats to biodiversity and why it doesn't matter. Nat Commun 13, 2616 (2022).

4.3. These latter points raised a question in my mind. The dataset includes a large fraction (roughly 1/3) of experimental studies (perhaps a higher fraction for climate change?). This in itself is a strength, but it also raises some concerns of possible experimental artefacts. It has long been known, for instance, that experimental studies on plant species invasions provide contrasting results when compared to observational studies. In part this may be because experiments tend to be performed at much finer spatial scales than many observational studies (although spatial grain is one of the issues analysed here). However, a second concern is that experimental plots are frequently isolated from one another by dissimilar habitat, a feature which might increase beta diversity between plots and thus reduce biotic homogeneity. It would be helpful if the authors could differentiate experimental and observational studies in their analysis (at least in the extended data section).

> We agree with you that combining experimental and field data is a strength, but can also lead to certain biases. To assess the differences between observational and experimental studies, we followed your recommendation and conducted a stratified analysis by splitting the dataset and replicating our approach on each group. The results are presented in Supplementary Information S1 and S2. These additional analyses show that the global analysis primarily reflects the observational data (which make up $\frac{2}{3}$ of the global dataset). It is interesting to note that generally the analysis of the experimental data produces conclusions in the same direction as the analysis of the observational data and the global analysis, but with greater uncertainty (wider confidence intervals, probably due to a smaller number of observations per subgroup).

Manuscript L.161–165

Yet, we acknowledge that experimental studies of these two human pressures may have compared reference controls to generally relatively high treatment levels (see Supplementary Information sections S1 and S2 for stratified analyses separating experimental from observational data) and that ranking human pressures can be strongly context and metric dependent³⁷.

[FIGURE REDACTED]

Figure S2 Impacts of human pressures on homogeneity and shifts in composition of biological communities estimated from observational studies. a, Log-response ratio of community homogeneity (logarithm-transformed ratio of impacted to reference values, LRR homogeneity). b, Log-response ratio of community composition shift (LRR shift). The global response (all data) is shown on the first row of each panel and is separated by factors in the following rows. The numbers between parentheses indicate the number of comparisons. For each category the dot represents the marginal mean computed from the model; dot size is proportional to number of studies included. The larger bar shows the 95 % confidence interval and the thinner bar represents the 99 % confidence interval.

[FIGURE REDACTED]

Figure S3 Impacts of human pressures on local diversity estimated from observational studies. Log-response ratio of local diversity (logarithm-transformed ratio of impacted to reference values, LRR local diversity). The global response (all data) shown on the first row is separated by factors in the following rows. The numbers between parentheses indicate the number of comparisons. For each category the dot represents the marginal mean computed from the model; dot size is proportional to number of studies included. The larger bar shows the 95 % confidence interval and the thinner bar represents the 99 % confidence interval.

[FIGURE REDACTED]

Figure S5 Impacts of human pressures on homogeneity and shifts in composition of biological communities estimated from experimental studies. a, Log-response ratio of community homogeneity (logarithm-transformed ratio of impacted to reference values, LRR homogeneity). b, Log-response ratio of community composition shift (LRR shift). The global response (all data) is shown on the first row of each panel and is separated by factors in the following rows. The numbers between parentheses indicate the number of comparisons. For each category the dot represents the marginal mean computed from the model; dot size is proportional to number of studies included. The larger bar shows the 95 % confidence interval and the thinner bar represents the 99 % confidence interval.

[FIGURE REDACTED]

Figure S6 Impacts of human pressures on local diversity estimated from experimental studies. Log-response ratio of local diversity (logarithm-transformed ratio of impacted to reference values, LRR local diversity). The global response (all data) shown on the first row is separated by factors in the following rows. The numbers between parentheses indicate the number of comparisons. For each category the dot represents the marginal mean computed from the model; dot size is proportional to number of studies included. The larger bar shows the 95 % confidence interval and the thinner bar represents the 99 % confidence interval.

4.4. Overall, I found this an interesting paper, and one making a potentially useful contribution. Nonetheless, it requires a bit more methodological clarity and circumspection before publication in a journal of this calibre could be warranted.

> We thank you once again for your supportive and constructive review and we are confident that this revision will meet all your expectations!

Authors' Response to Reviews of The global human impact on biodiversity

F. Keck *et al.*

Submitted to *Nature* [2023-11-21089A]

Revision 2

Referee #1

I would like to congratulate the authors for the tremendous work they did in replying to my comments on the previous draft of the manuscript. It was truly a pleasure to read the reply. I only have one very minor comment below:

> Thank you very much for your kind words and for your thoughtful feedback throughout the review process. We're glad to hear that our revisions have addressed your comments effectively. We appreciate your additional suggestion and have incorporated it into the manuscript as recommended (see below).

(1) You state "As PCoA and NMDS are directly derived from pairwise beta-diversity matrices," (l.254-259). Is this always the case, this is, you only use PCoA and NDMS from pairwise distance matrices or in some cases the PCoA where done directly on the original community matrices?

> Thank you for your observation, indeed, PCoA and NMDS are always derived from pairwise beta-diversity matrices. This is actually not a choice on our part, but the direct consequence of the mathematical definition of PCoA that does not allow the use of anything other than a symmetrical distance matrix as an input. This is reflected in the algorithm's software implementation. For example, R's *cmdscale* and Python's *skbio.stats.ordination.pcoa* commands, which both implement PCoA, accept nothing other than a distance matrix. Commands that offer the option of supplying a raw data matrix (e.g. *capscale* of the *vegan* package) actually calculate the dissimilarity matrix from the raw data under the hood. Similarly, NMDS requires a dissimilarity matrix, as it is inherently rank-based and cannot be applied directly to raw data matrices of abundances or counts. We clarified this point in the relevant section.

Manuscript L.258–259

To be included a publication had to feature at least one plot reporting the projection of biological communities in the first two dimensions of a PCoA or NMDS analysis computed from a community dissimilarity matrix, since both of these methods strictly require a distance matrix as input, and are therefore preferred for the graphical representation of beta-diversity.

I look forward to see this paper in print.

> As we do! Thank you.

Referee #2

I appreciate a very thorough revision of the ms made by the authors. All my comments have been addressed, in particular new sensitivity analyses have been conducted to test how results from unweighted models differed from those of the weighted models and publication bias tests have been conducted. Other methodological details concerning the meta-analysis have been clarified as well. I think the ms has greatly improved as a result of revision.

> Thank you very much for your positive feedback and for recognizing the efforts we made in revising the manuscript. We appreciate your insights and recommendations, which significantly contributed to improving the clarity and robustness of our analysis. We are pleased to hear that the additional sensitivity analyses and methodological clarifications have addressed your comments and enhanced the manuscript.

Referee #4

I was positively impressed with this manuscript when I first reviewed it, and the revised manuscript is substantially improved, with clearer and more informative prose (especially as to methodologies). All of my original substantive concerns have been addressed adequately.

> Thank you for your positive feedback and for recognizing the improvements in clarity and detail. We are pleased that the revisions addressed your previous comments effectively. Below, we address in detail the few additional points you have raised.

A few additional points:

(a) The order in which the different sets of results are presented shifts over the course of the paper. In my mind, the "logical" order would be first diversity ("D": how many types), then composition ("C": WHICH types), then homogenisation ("H": similarity between samples in which types), but that is seldom used here. Instead it is sometimes CHD (lines 42-43), sometimes CDH (lines 44-46), sometimes DCH (lines 55-59), and then ultimately HCD (lines 110-114 and thereafter). In the methodology section (lines 308-326) it shifts again: back to CHD. This isn't important, but it is sometimes a bit confusing.

> Thank you very much, this is very valuable feedback. Indeed we realise—also when having presented our results at scientific conferences and in discussion with colleagues—that there are two interlinked levels of complexity that came up repeatedly and caused some confusion. Firstly about the “logic” order of presenting and secondly about the different measures of local diversity. Especially the latter was often perceived as unclear, as we had summarised different metrics (richness, Shannon etc.), while most people think in richness terms. This was also pointed out in the previous round of review (comments #1.4, #3.6, #4.1). The order of presentation itself was a mix by integrating how we present our findings vs. how the field as such has studied (sequentially and over time) richness/diversity,

composition and then homogenization. That is, the classic development of the field was likely DCH, while the more current focus is HCD.

We have now addressed this in the following way: firstly, we clarified the structure of order of presentation. We generally aim to present in the HCD order when discussing our results, as this reflects the focus of the manuscript (homogenization and composition being the focal motivation; and also diversity being a subset of the data only, the latter is one reason why we do not start with diversity). This is the order of the figures too. Yet, in the abstract and when reflecting past work, we maintain the DCH order, as this is the order in which the field developed and we think people are most familiar with (as also highlighted by the two other reviewers, which were supportive of this structure). By this, we best reflect past work and the focus of our own work, respectively. Secondly, we have simplified the diversity to only reflect the most common metric of species richness. Doing so simplifies our message and does not qualitatively or quantitatively change the results.

Manuscript L.42

For all comparisons, we quantified three key measures of biodiversity to assess how these human pressures drive **homogenisation and shifts in composition** of biological communities across space and changes in local diversity, respectively.

Manuscript L.238

To detect studies reporting results of **homogenisation and compositional shift** in response to human pressures

Manuscript L.315

We measured the magnitude of **homogenisation and compositional shift** under human pressure, across all the studies.

Manuscript L.319–328

First, a measurement of the change in community similarity estimated as the log-transformed ratio of the mean distance within impacted communities D_I and the mean distance within reference communities D_R . For clarity, this value is multiplied by -1 to express a change in homogeneity (LRR homogeneity) instead of heterogeneity, Eq. (1). Second, a measurement of the shift in community composition (LRR shift) estimated as the log-transformed ratio of the mean distance between impacted and reference communities D_B and the mean distance within impacted and reference communities D_W , Eq. (2).

(1) LRR homogeneity = $-\ln(DI/DR)$

(2) LRR shift = $\ln(DB/DW)$

Manuscript L.332–333

Our results show that LRR homogeneity and LRR shift are generally robust to the inclusion of additional groups in the PCoA and NMDS graphs from which they are derived.

Manuscript L.336

For each record, the mean local diversity for the impacted and reference communities was also recorded (if reported as taxonomic richness).

(b) The presentation of the LRR homogeneity results seems peculiar, with verbal interpretations given for very NON-significant trends (e.g. for effects of Invasive spp, line 135), whereas the strongly significant (de-homogenising) effects of drivers such as resource exploitation are not mentioned. The fact that LRR homogenisation interacts with spatial scale is particularly interesting, but the text (lines 128-130) gives equal emphasis to the consistent and significant de-homogenising patterns found at fine spatial scales and the inconsistent and non-significant homogenising patterns at coarse scales. It also doesn't mention that these scaling properties are quite different for the experimental versus observational studies, a fact that might shed light on the discussion of stochasticity and drift (lines 140-142), as well as on the inference of causality (lines 212-213).

> Thank you, we happily revised the presentation of significant vs. non-significant results, expanding on the former. We agree that the strong, de-homogenizing effects should receive adequate attention in our discussion. Based on your suggestions, we specifically removed the reference to the invasive species results as indeed this is not a main line of support to back our claim. Instead, we now discuss in this paragraph the strongest and statistically significant effects on biotic differentiation. This includes, as also pointed out and recommended by you, resource exploitation and pollution, which both have also very direct real-world implications. We see that these changes offer a more balanced presentation and discussion of results, and better reflect our main findings.

Manuscript L.142–146

Finally, stochastic effects and ecological drift can promote biotic differentiation, and are likely to play more important roles in local impact studies where strong pressures can completely destabilise communities by drastically reducing the number of individuals. In our systematic analysis, we indeed find a significant biotic differentiation in response to resource exploitation (LRR homogeneity = -0.117, 95% CI = -0.197 to -0.036) and

pollution (LRR homogeneity = -0.071, 95% CI = -0.129 to -0.012), two types of human pressure capable of modifying ecosystems in a pronounced way over a short period of time, and thus increasing the importance of ecological drift in community assembly.

(c) I am left slightly mystified by the interpretation of the "LRR shift" results. If I understand correctly, these refer to log response ratios in community compositional differences ("shifts") between impacted and reference plots, relative those found within the sets of impacted and reference communities (see lines 308-313). This comparison of variance between groups to variance within groups is familiar (e.g. in ANOVA), but it includes no direct measure of community changes over time relative to some baseline (community "shifts" per se), as to do so the data would need to include prior community compositions (as in a BACI design, which the paper makes clear in lines 197-202 is NOT used here). Thus so long as the treatment groups differ from each other, the "shift" will be attributed to the "impacted" group rather than the reference group. I am also somewhat confused about the appropriate degree of statistical confidence to claim in analyses of this sort. A set of N values will have $(N^2 - N)/2$ unique within-group non-self comparisons, and those between treatments will have $(N_1 \times N_2)/2$ unique comparisons. These comparisons are not fully independent, as e.g. a single highly unusual community creates a large number of long distance values. Surely there are specialised methods (e.g. Mantel tests) for distance matrices of this sort?

> Thank you for your comment, we are happy to clarify our approach. Regarding the interpretation of "LRR shift" results, you are correct that the shift is derived from comparisons between impacted and reference communities. This is based on a commonly used assumption (also generally reflecting a "naive" yet realistic expectation) that the reference set functions as a baseline for detecting impact-related changes (yet, importantly, without giving a qualitative/normative statement on the direction of this change, thus the "impact" does not per se have to be "negative", it is more generally reflecting a "original/previous" vs. a "derived" state). This approach implicitly assumes that our sampled communities within the impacted and reference groups are comparable, with the reference group providing a stable baseline.

Given that all of the comparisons made were already directly given/derived from unique original studies that had explicitly described them, this comparison is valid and justifiable, as we must rely on the original studies' judgement about direct comparability. Given that our study design does not include a direct temporal baseline as in a BACI approach, this interpretation may simplify certain dynamic aspects of community change, yet it is not fundamentally affecting the control vs. impact understanding in a general term. We have now noted this assumption more explicitly in the revised manuscript.

Manuscript L.201–204

We circumvent these challenges by systematically comparing impacted sites with reference sites (i.e., "Control-Impact" design). In such designs, the control and impacted sets of sites are assumed to be comparable and any differences between the two treatments is attributed to a change in the impacted group relative to the reference

group, which is considered as a stable baseline. While this approach may be in some cases less sensitive than the gold-standard “Before-After Control-Impact” (BACI) design that explicitly accounts for pre-existing differences between the impact and reference groups⁴⁶, it remains by far the most widely used method to measure the real and direct effect of human pressures on biological communities (>95 % of the studies considered had a “Control-Impact” design, and <5% had a BACI design). With this impact-focused perspective, we quantify and recall the direct and adverse effects of human pressures on local biodiversity.

On the statistical confidence and independence of distance values, we understand the comment regarding the potential for overestimating confidence due to the non-independence of within-group comparisons. Indeed, comparisons within and between groups can yield a large number of non-independent values, which could theoretically inflate variance estimates if treated as independent. Importantly, however, this was not the case and not problematic in our study: our primary aim is to derive effect sizes based on community compositional differences, rather than conducting statistical tests within individual studies.

We have now emphasised this aspect and clarified the text relative to your comment, to avoid misunderstandings. These effect sizes are then synthesised at the inter-study level, where statistical procedures account for variance across multiple studies. We thus have no formal statistical testing on distance matrices within individual studies, as this would indeed require specialised methods (e.g., Mantel tests or alternative approaches for handling distance-based non-independence) that were not the intended scope of our current analysis. We added a sentence in the Method section to clarify this important point.

Manuscript L.341–344

The three effect sizes described here were not statistically tested individually (i.e. within-study statistical testing). Statistical modelling and testing was instead conducted across studies, where variability was accounted for in inter-study synthesis, as described in the next section.

Overall, however, I find the paper much improved, and expect it will make a strong contribution to the growing literature on human biodiversity impacts. I am happy to recommend its publication, once any remaining (mostly minor) issues are addressed.

> Thank you for your constructive feedback and your thoughtful and supportive comments! We are very pleased that you find the revisions have strengthened the paper and we appreciate your recommendation for publication. We have carefully addressed each of the remaining comments stated above and we are confident these changes fully meet your expectations.